# APPROXIMATION AND NON-PARAMETRIC ESTIMATION OF RESNET-TYPE CONVOLUTIONAL NEURAL NETWORKS VIA BLOCK-SPARSE FULLY-CONNECTED NEURAL NETWORKS

## ABSTRACT

We develop new approximation and statistical learning theories of convolutional neural networks (CNNs) via the ResNet-type structure where the channel size, filter size, and width are fixed. It is shown that a ResNet-type CNN is a universal approximator and its expression ability is no worse than fully-connected neural networks (FNNs) with a *block-sparse* structure even if the size of each layer in the CNN is fixed. Our result is general in the sense that we can automatically translate any approximation rate achieved by block-sparse FNNs into that by CNNs. Thanks to the general theory, it is shown that learning on CNNs satisfies optimality in approximation and estimation of several important function classes.

As applications, we consider two types of function classes to be estimated: the Barron class and Hölder class. We prove the clipped empirical risk minimization (ERM) estimator can achieve the same rate as FNNs even the channel size, filter size, and width of CNNs are constant with respect to the sample size. This is minimax optimal (up to logarithmic factors) for the Hölder class. Our proof is based on sophisticated evaluations of the covering number of CNNs and the non-trivial parameter rescaling technique to control the Lipschitz constant of CNNs to be constructed.

## 1 INTRODUCTION

Convolutional Neural Network (CNN) is one of the most popular architectures in deep learning research, with various applications such as computer vision (Krizhevsky et al. (2012)), natural language processing (Wu et al. (2016)), and sequence analysis in bioinformatics (Alipanahi et al. (2015), Zhou & Troyanskaya (2015)). Despite practical popularity, theoretical justification for the power of CNNs is still scarce from the viewpoint of statistical learning theory.

For fully-connected neural networks (FNNs), there is a lot of existing work, dating back to the 80's, for theoretical explanation regarding their *approximation* ability (Cybenko (1989), Barron (1993), Lu et al. (2017), Yarotsky (2017), and Petersen & Voigtlaender (2017)) and *generalization* power (Barron (1994), Arora et al. (2018), and Suzuki (2018)). See also Pinkus (2005) and Kainen et al. (2013) for surveys of earlier works. Although less common compared to FNNs, recently, statistical learning theory for CNNs has been studied, both about approximation ability (Zhou (2018), Yarotsky (2018), Petersen & Voigtlaender (2018)) and about generalization power (Zhou & Feng (2018)). One of the standard approaches is to relate the approximation ability of CNNs with that of FNNs, either deep or shallow. For example, Zhou (2018) proved that CNNs are a universal approximator of the Barron class (Barron (1993), Klusowski & Barron (2016)), which is a historically important function class in the approximation theory. Their approach is to approximate the function using a 2-layered FNN (i.e., an FNN with a single hidden layer) with the ReLU activation function (Krizhevsky et al. (2012)) and transform the FNN into a CNN. Very recently independent of ours, Petersen & Voigtlaender (2018) showed any function realizable with an FNN can extend to an equivariant function realizable by a CNN that has the same order of parameters. However, to the best of our knowledge, no CNNs that achieves the minimax optimal rate (Tsybakov (2008), Giné & Nickl (2015)) in important function classes, including the Hölder class, can keep the number of

units in each layer constant with respect to the sample size. Architectures that have extremely large depth, while moderate channel size and width have become feasible, thanks to recent methods such as identity mappings (He et al. (2016), Huang et al. (2018)), sophisticated initialization schemes (He et al. (2015), Chen et al. (2018)), and normalization techniques (Ioffe & Szegedy (2015), Miyato et al. (2018)). Therefore, we would argue that there are growing demands for theories which can accommodate such constant-size architectures.

In this paper, we analyze the learning ability of ResNet-type ReLU CNNs which have identity mappings and constant-width residual blocks with fixed-size filters. There are mainly two reasons that motivate us to study this type of CNNs. First, although ResNet is the de facto architecture in various practical applications, the approximation theory for ResNet has not been explored extensively, especially from the viewpoint of the relationship between FNNs and CNNs. Second, constant-width CNNs are critical building blocks not only in ResNet but also in various modern CNNs such as Inception (Szegedy et al. (2015)), DenseNet (Huang et al. (2017)), and U-Net (Ronneberger et al. (2015)), to name a few. Our strategy is to replicate the learning ability of FNNs by constructing tailored ResNet-type CNNs. To do so, we pay attention to the *block-sparse* structure of an FNN, which roughly means that it consists of a linear combination of multiple (possibly dense) FNNs (we define it rigorously in the subsequent sections). Block-sparseness decreases the model complexity coming from the combinatorial sparsity patterns and promotes better bounds. Therefore, it is often utilized, both implicitly or explicitly, in the approximation and learning theory of FNNs (e.g., Bölcskei et al. (2017), Yarotsky (2018)). We first prove that if an FNN is block-sparse with $M$ blocks ($M$-way block-sparse FNN), we can realize the FNN with a ResNet-type CNN with $O(M)$ additional parameters, which are often negligible since the original FNN already has $\Omega(M)$ parameters. Using this approximation, we give the upper bound of the estimation error of CNNs in terms of the approximation errors of block sparse FNNs and the model complexity of CNNs. Our result is general in the sense that it is not restricted to a specific function class, as long as we can approximate it using block-sparse FNNs.

To demonstrate the wide applicability of our methods, we derive the approximation and estimation errors for two types of function classes with the same strategy: the Barron class (of parameter $s = 2$) and Hölder class. We prove, as corollaries, that our CNNs can achieve the approximation error of order $\tilde{O}(M^{-\frac{D+2}{2D}})$ for the Barron class and $\tilde{O}(M^{-\frac{\beta}{D}})$ for the $\beta$-Hölder class and the estimation error of order $\tilde{O}_p(N^{-\frac{D+2}{2(D+1)}})$ for the Barron class and $\tilde{O}_p(N^{-\frac{2\beta}{2\beta+D}})$ for the $\beta$-Hölder class, where $M$ is the number of parameters (we used $M$ here, same as the number of blocks because it will turn out that CNNs have $O(M)$ blocks for these cases), $N$ is the sample size, and $D$ is the input dimension. These rates are same as the ones for FNNs ever known in the existing literature. An important consequence of our theory is that the ResNet-type CNN can achieve the minimax optimal estimation error (up to logarithmic factors) for $\beta$-Hölder class even if its filter size, channel size and width are constant with respect to the sample size, as opposed to existing works such as Yarotsky (2017) and Petersen & Voigtlaender (2018), where optimal FNNs or CNNs could have a width or a channel size goes to infinity as $N \to \infty$.

In summary, the contributions of our work are as follows:

- We develop the approximation theory for CNNs via ResNet-type architectures with constant-width residual blocks. We prove any $M$-way block-sparse FNN is realizable such a CNN with $O(M)$ additional parameters. That means if FNNs can approximate a function with $O(M)$ parameters, we can approximate the function with CNNs at the same rate (Theorem 1).

- We derive the upper bound of the estimation error in terms of the approximation error of FNNs and the model complexity of CNNs (Theorem 2). This result gives the sufficient conditions to derive the same estimation error as that of FNNs (Corollary 1).

- We apply our general theory to the Barron class and Hölder class and derive the approximation (Corollary 2 and 4) and estimation (Corollary 3 and 5) error rates, which are identical to those for FNNs, even if the CNNs have constant channel and filter size with respect to the sample size. In particular, this is minimax optimal for the Hölder case.

|  | Zhou (2018) | Petersen & Voigtlaender (2018) | Ours |
|---|---|---|---|
| CNN type | Conventional | Conventional | ResNet |
| Function type | Barron ($s = 2$) | Any (FNNs) | Any (block-sparse FNNs) |
| Channel size (Dense FNN case) | 1 | $\geq 1$ | $\geq 1$ |
| Channel size ($\beta$-Hölder case) | N.A. | $\tilde{O}(\varepsilon^{-\frac{D}{\beta}})$ | $O(1)$ |
| Width | Increasing | Fixed | Fixed |
| Filter size | Fixed | Full | Fixed |
| Norm bound | No | Yes | Yes |
| Padding | Yes | No | Yes |

Table 1: Comparison of CNN architectures. "Channel size (Dense FNN case)": The number of channels needed to realize a function represented by a fixed-width dense FNN. "Channel size ($\beta$-Hölder case)": The number of channles needed to approximate a $\beta$-Hölder function with accuracy $\varepsilon$ measured by the sup norm. "Increasing": The width of layer is monotonically increasing. "Full": Filter size is as large as the layer width. "Padding": Whether the theory includes convolution operations with padding.

## 2 RELATED WORK

We summarize in Table 1 the differences in the CNN architectures between our work and Zhou (2018) and Petersen & Voigtlaender (2018), which established the approximation theory of CNNs via FNNs. First and foremost, Zhou (2018) only considered a specific function class — the Barron class — as a target function class, although their method is applicable to any function class that can be realized by a 2-layered ReLU FNN. Regarding the architecture, they considered CNNs with a single channel and whose width is "linearly increasing" (Zhou (2018)) layer by layer. For regression or classification problems, it is rare to use such an architecture. In addition, since they did not bound the norm of parameters in the approximating CNNs, we cannot derive the estimation error from this method. Petersen & Voigtlaender (2018) fully utilized the group invariance structure of underlying input spaces to construct CNNs. Such a structure makes theoretical analysis easier, especially for investigating the equivariance properties of CNNs since it enables us to incorporate mathematical tools such as group theory, Fourier analysis, and representation theory. Although their results are quite strong in that it is applicable to any function that can be approximated by FNNs, their assumption on the group structure excludes the padding convolution layer, an important and popular type of convolution operations. Another point is that if we simply apply their construction method to derive the estimation error for (equivariant) Hölder functions, combined with the approximation result of Yarotsky (2017), the resulting CNN that achieves the minimax optimal rate has $\tilde{O}(\varepsilon^{-\frac{D}{\beta}})$ channels where $\varepsilon$ is the approximation error threshold. It is partly because their construction is not aware of the internal sparse structure of approximating FNNs. Finally, the filter size of their CNN is as large as the input dimension. As opposed to these two works, we employ padding- and ResNet-type CNNs which have multiple channels, fixed-size filters, and constant widths. Like Petersen & Voigtlaender (2018), our result is applicable to any function, as long as the FNNs to be approximated are block sparse, including the Barron and Hölder cases. If we apply our theorem to these classes, we can show that the optimal CNNs can achieve the same approximation and estimation rate as FNNs, while the number of channels is independent of the sample size. Further, this is minimax optimal up to the logarithmic factors for the Hölder class.

Due to its practical success, theoretical analysis for ResNet has been explored recently (e.g., Lin & Jegelka (2018), Lu et al. (2018), Nitanda & Suzuki (2018), and Huang et al. (2018)). From the viewpoint of statistical learning theory, Nitanda & Suzuki (2018) and Huang et al. (2018) investigated the generalization power of ResNet from the perspective of the boosting interpretation. However, they did not discuss the function approximation ability of ResNet. To the best of our knowledge, our theory is the first work to provide the approximation ability of the CNN class that can accommodate the ResNet-type ones.

We import the approximation theories for FNNs, especially ones for the Barron class and Hölder class. The approximation theory for the Barron class has been investigated in e.g., Barron (1993), Klusowski & Barron (2016), and Lee et al. (2017). Originally Barron (1993) considered the parameter $s = 1$ (see Definition 3) and the activation function $\sigma$ satisfying $\sigma(z) \to 1$ as $z \to \infty$ and $\sigma(z) \to 0$ as $z \to -\infty$. Later, Klusowski & Barron (2016) studied the approximation theory with $s = 2$ and proved that 2-layered ReLU FNNs with $M$ hidden units can approximate functions of this class with the order of $\tilde{O}(M^{-\frac{D+2}{2D}})$. Yarotsky (2017) proved FNNs with $O(S)$ non-zero parameters can approximate $\beta$-Hölder continuous functions with the order of $\tilde{O}(S^{-\frac{\beta}{D}})$. Using this bound, Schmidt-Hieber (2017) proved that the estimation error of the ERM estimator is $\tilde{O}(N^{-\frac{2\beta}{2\beta+D}})$, which is minimax optimal up to logarithmic factors (see, e.g., Tsybakov (2008)).

## 3 PROBLEM SETTING

### 3.1 EMPIRICAL RISK MINIMIZATION

We consider a regression task in this paper. Let $X$ be a $[-1, 1]^D$-valued random variable with unknown probability distribution $\mathcal{P}_X$ and $\xi$ be an independent random noise drawn from the Gaussian distribution with an unknown variance $\sigma^2$: $\xi \sim \mathcal{N}(0, \sigma^2)$ ($\sigma > 0$). Let $f^\circ$ be an unknown deterministic function $f^\circ : [-1, 1]^D \to \mathbb{R}$ (we will characterize $f^\circ$ rigorously in the theorems later). We define a random variable $Y$ by $Y := f^\circ(X) + \xi$. We denote the joint distribution of $(X, Y)$ by $\mathcal{P}$. Suppose we are given a dataset $\mathcal{D} = ((x_1, y_1), \ldots, (x_N, y_N))$ independently and identically sampled from the distribution $\mathcal{P}$, we want to estimate the true function $f^\circ$ from the finite dataset $\mathcal{D}$.

We evaluate the performance of an estimator by the squared error. For a measurable function $f : [-1, 1]^D \to \mathbb{R}$, we define the *empirical error* of $f$ by $\hat{\mathcal{R}}_\mathcal{D}(f) := \sum_{n=1}^N (y_n - f(x_n))^2$ and the *estimation error* by $\mathcal{R}(f) := \mathbb{E}_{X,Y} \left[ (f(X) - Y)^2 \right]$. Given a subset $\mathcal{F}$ of measurable functions from $[-1, 1]^D \to \mathbb{R}$, we consider the *clipped empirical risk minimization (ERM) estimator* $\hat{f}$ of $\mathcal{F}$ that satisfies

$$\hat{f} := \text{clip}[f_{\min}] \quad \text{where } f_{\min} \in \underset{f \in \mathcal{F}}{\arg\min} \, \hat{\mathcal{R}}_\mathcal{D}(\text{clip}[f]).$$

Here, $\text{clip}$ is the clipping operator defined by $\text{clip}[f] := (f \vee -\|f^\circ\|_\infty) \wedge \|f^\circ\|_\infty$. For a measurable function $f : [-1, 1]^D \to \mathbb{R}$, we define the $L_2$-norm (weighted by $\mathcal{P}_X$) and the sup norm of $f$ by $\|f\|_{\mathcal{L}^2(\mathcal{P}_X)} := \left( \int_{[-1,1]^D} f^2(x) \mathrm{d}\mathcal{P}_X(x) \right)^{\frac{1}{2}}$ and $\|f\|_\infty := \sup_{x \in [-1,1]^D} |f(x)|$, respectively. Let $\mathcal{L}^2(\mathcal{P}_X)$ be the set of measurable functions $f$ such that $\|f\|_{\mathcal{L}^2(\mathcal{P}_X)} < \infty$ with the norm $\| \cdot \|_{\mathcal{L}^2(\mathcal{P}_X)}$. The task is to estimate the *approximation* error $\min_{f \in \mathcal{F}} \|f - f^\circ\|_\infty$ and the *estimation* error of the clipped ERM estimator: $\mathcal{R}(\hat{f}) - \mathcal{R}(f^\circ)$. Note that the estimation error is a random variable with respect the choice of the training dataset $\mathcal{D}$. By the definition of $\mathcal{R}$ and the independence of $X$ and $\xi$, the estimation error equals to $\|\hat{f} - f^\circ\|_{\mathcal{L}^2(P_X)}^2$.

### 3.2 CONVOLUTIONAL NEURAL NETWORKS

In this section, we define CNNs used in this paper. For this purpose, it is convenient to introduce $\ell_0$, the set of real-valued sequences whose finitely many elements are non-zero: $\ell_0 := \{w = (w_n)_{n \in \mathbb{N}_{>0}} \mid \exists N \in \mathbb{N}_{>0} \text{ s.t. } w_n = 0, \forall n \geq N\}$. $w = (w_1, \ldots, w_K) \in \mathbb{R}^K$ can be regarded as an element of $\ell_0$ by setting $w_n = 0$ for all $n > K$. Likewise, for $C, C' \in \mathbb{N}_{>0}$, which will be the input and output channel sizes, respectively, we can think of $(w_{k,j,i})_{k \in [K], j \in [C'], i \in [C]} \in \mathbb{R}^{K \times C' \times C}$ as an element of $\ell_0^{C' \times C}$. For a filter $w = (w_{n,j,i})_{n \in \mathbb{N}_{>0}, i \in [C], j \in [C']} \in \ell_0^{C' \times C}$, we define the *one-sided padding and stride-one convolution* by $w$ as an order-4 tensor $L_D^w = ((L_D^w)_{\alpha,i}^{\beta,j}) \in \mathbb{R}^{D \times D \times C' \times C}$ by

$$(L_D^w)_{\alpha,i}^{\beta,j} := \begin{cases} w_{(\alpha-\beta+1),j,i} & \text{if } 0 \leq \alpha - \beta \leq D - 1 \\ 0 & \text{otherwise.} \end{cases}$$

Here, $i$ (resp. $j$) runs through 1 to $C$ (resp. $C'$) and $\alpha$ and $\beta$ runs through 1 to $D$. Since we fix the input dimension $D$ throughout the paper, we will omit the subscript $D$ and write as $L^w$ if it is obvious from context.

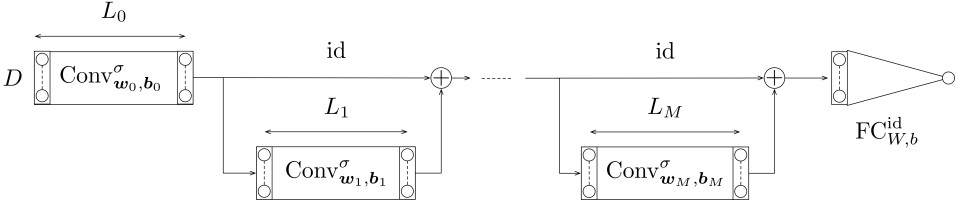

Figure 1: ResNet-type CNN defined in Definition 1. Variables are as in Definition 1.

**Remark 1.** *For $K \le K'$, we can embed $\mathbb{R}^K$ into $\mathbb{R}^{K'}$ by inserting zeros: $w = (w_1, \dots, w_K) \mapsto w' = (w_1, \dots, w_K, 0, \dots, 0)$. It is easy to show $L^w = L^{w'}$. Using this equality, we can expand a size-$K$ filter to size-$K'$.*

We can interpret $L^w$ as a linear mapping from $\mathbb{R}^{D \times C}$ to $\mathbb{R}^{D \times C'}$. Specifically, for $x = (x^{\alpha,i})_{\alpha,i} \in \mathbb{R}^{D \times C}$, we define $(y^{\beta,j})_{\beta,j} = L^w(x) \in \mathbb{R}^{D \times C'}$ by

$$y^{\beta,j} := \sum_{i,\alpha} (L^w)^{\beta,j}_{\alpha,i} \, x^{\alpha,i}.$$

Next, we define the building block of CNNs: convolutional layers and fully-connected layers. Let $C, C', K \in \mathbb{N}_{>0}$ be the input channel size, output channel size, and filter size, respectively. For a weight tensor $w \in \mathbb{R}^{K \times C' \times C}$, a bias vector $b \in \mathbb{R}^{C'}$, and an activation function $\sigma : \mathbb{R} \to \mathbb{R}$, we define the *convolutional layer* $\mathrm{Conv}^\sigma_{w,b} : \mathbb{R}^{D \times C} \to \mathbb{R}^{D \times C'}$ by $\mathrm{Conv}^\sigma_{w,b}(x) := \sigma(L^w(x) - \mathbf{1}_D \otimes b)$ where, $\otimes$ is the outer product of vectors and $\sigma$ is applied in element-wise manner. Similarly, let $W \in \mathbb{R}^{D \times C}$, $b \in \mathbb{R}$, and $\sigma : \mathbb{R} \to \mathbb{R}$, we define the *fully-connected layer* $\mathrm{FC}^\sigma_{W,b} : \mathbb{R}^{D \times C} \to \mathbb{R}$ by $\mathrm{FC}^\sigma_{W,b}(a) = \sigma(\mathrm{vec}(W)^\top \mathrm{vec}(a) - b)$. Here, $\mathrm{vec}(\cdot)$ is the vectorization operator that flattens a matrix into a vector.

Finally, we define the ResNet-type CNN as a sequential concatenation of one convolution block, $M$ residual blocks, and one fully-connected layer. Figure 1 is the schematic view of the CNN we adopt in this paper.

**Definition 1** (Convolutional Neural Networks (CNNs)). *Let $M \in \mathbb{N}_{>0}$ and $L_m \in \mathbb{N}_{>0}$, which will be the number of residual blocks and the depth of $m$-th block, respectively. Let $C_m^{(l)}, K_m^{(l)}$ be the channel size and filter size of the $l$-th layer of the $m$-th block for $m = 0, \dots, M$[1] and $l \in [L_m]$. We assume $C_0^{(L_0)} = C_1^{(L_1)} = \cdots = C_M^{(L_M)}$. Let $w_m^{(l)} \in \mathbb{R}^{K_m^{(l)} \times C_m^{(l)} \times C_m^{(l-1)}}$ and $b_m^{(l)} \in \mathbb{R}$ be the weight tensors and biases of $l$-th layer of the $m$-th block in the convolution part, respectively. Finally, let $W \in \mathbb{R}^{D \times C_0^{(L_0)}}$ and $b \in \mathbb{R}$ be the weight matrix and the bias for the fully-connected layer part, respectively. For $\boldsymbol{\theta} := ((w_m^{(l)})_{m,l}, (b_m^{(l)})_{m,l}, W, b)$ and an activation function $\sigma : \mathbb{R} \to \mathbb{R}$, we define $\mathrm{CNN}^\sigma_{\boldsymbol{\theta}} : \mathbb{R}^D \to \mathbb{R}^D$, the CNN constructed from $\boldsymbol{\theta}$, by*

$$\mathrm{CNN}^\sigma_{\boldsymbol{\theta}} := \mathrm{FC}^{\mathrm{id}}_{W,b} \circ (\mathrm{Conv}^\sigma_{\boldsymbol{w}_M, \boldsymbol{b}_M} + \mathrm{id}) \circ (\mathrm{Conv}^\sigma_{\boldsymbol{w}_{M-1}, \boldsymbol{b}_{M-1}} + \mathrm{id}) \circ \cdots$$
$$\circ (\mathrm{Conv}^\sigma_{\boldsymbol{w}_1, \boldsymbol{b}_1} + \mathrm{id}) \circ \mathrm{Conv}^\sigma_{\boldsymbol{w}_0, \boldsymbol{b}_0},$$

*where $\mathrm{Conv}^\sigma_{\boldsymbol{w}_m, \boldsymbol{b}_m} := \mathrm{Conv}^{\mathrm{id}}_{w_m^{(L_m)}, b_m^{(L_m)}} \circ \mathrm{Conv}^\sigma_{w_m^{(L_m-1)}, b_m^{(L_m-1)}} \circ \cdots \circ \mathrm{Conv}^\sigma_{w_m^{(1)}, b_m^{(1)}}$ and $\mathrm{id} : \mathbb{R}^{D \times C_0^{(L_0)}} \to \mathbb{R}^{D \times C_0^{(L_0)}}$ is the identity function.*

Although $\mathrm{CNN}^\sigma_{\boldsymbol{\theta}}$ in this definition has a fully-connected layer, we refer to the stack of convolutional layers both with or without the final fully-connect layer as a CNN in this paper. We say a *linear* convolutional layer or a *linear* CNN when the activation function $\sigma$ is the identity function and a *ReLU* convolution layer or a *ReLU* CNN when $\sigma$ is ReLU defined by $\mathrm{ReLU}(x) = x \vee 0$. We borrow the term from ResNet and call $\mathrm{Conv}^\sigma_{\boldsymbol{w}_m, \boldsymbol{b}_m}$ ($m > 0$) and id in the above definition the $m$-th *residual block* and the $m$-th identity mapping, respectively. We say a 4-tuple $\boldsymbol{\theta}$ is *compatible* with $(C_m^{(l)})_{m,l}$ and $(K_m^{(l)})_{m,l}$ when each component of $\boldsymbol{\theta}$ satisfies the aforementioned dimension conditions.

---

[1]Note that $m$ starts from 0. It is convenient for our purpose.

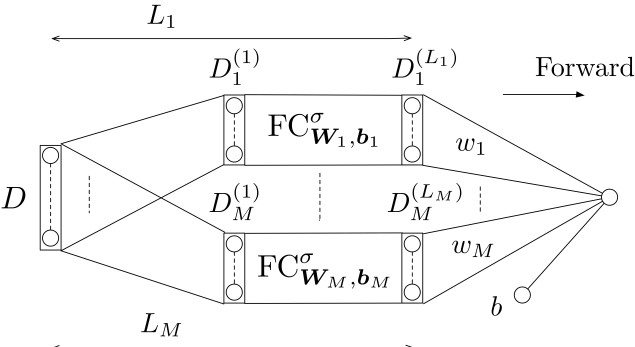

Figure 2: Schematic view of a block-sparse FNN. Variables are as in Definition 2.

For architecture parameters $\boldsymbol{C} = (C_m^{(l)})_{m,l}$ and $\boldsymbol{K} = (K_m^{(l)})_{m,l}$ ($m = 0, \ldots, M, l \in [L_m]$), and norm parameters for convolution layers $B^{(\mathrm{conv})} > 0$ and for fully-connected layers $B^{(\mathrm{fc})} > 0$, we define $\mathcal{F}^{(\mathrm{CNN})} = \mathcal{F}^{(\mathrm{CNN})}_{\boldsymbol{C},\boldsymbol{K},B^{(\mathrm{conv})},B^{(\mathrm{fc})}}$, the hypothesis class consisting of ReLU CNNs, as follows:

$$\mathcal{F}^{(\mathrm{CNN})} := \left\{ \mathrm{CNN}_{\boldsymbol{\theta}}^{\mathrm{ReLU}} \;\middle|\; \begin{array}{l} \boldsymbol{\theta} = ((w_m^{(l)})_{m,l}, (b_m^{(l)})_{m,l}, W, b) \text{ is compatible with } (\boldsymbol{C}, \boldsymbol{K}), \\ \max_{m=0,\ldots,M,l\in[L_m]} \|w_m^{(l)}\|_\infty \vee \|b_m^{(l)}\|_\infty \leq B^{(\mathrm{conv})}, \\ \|W\|_\infty \vee \|b\|_\infty \leq B^{(\mathrm{fc})} \end{array} \right\}.$$

Here, the domain of CNNs is restricted to $[-1, 1]^D$. Note that we impose norm constraints to the convolution part and fully-connected part separately. We emphasize that we do not impose any sparse constraints (e.g., restricting the number of non-zero parameters in a CNN to some fixed value) to $\mathcal{F}^{(\mathrm{CNN})}$, as opposed to previous literature such as Yarotsky (2017), Schmidt-Hieber (2017), and Imaizumi & Fukumizu (2018). Since the notation is cluttered, we sometimes omit the subscripts as we do in the above.

**Remark 2.** *In this paper, we adopted one-sided padding, which is not often used practically, in order to make proofs simple. However, with slight modifications, all statements are true for equally-padded convolutions, the widely employed padding style which adds (approximately) same numbers of zeros to both ends of an input signal, with the exception that the filter size $K$ is restricted to $K \leq \lfloor \frac{D}{2} \rfloor$ instead of $K \leq D - 1$. We also discuss our design choice, especially the comparison with the original ResNet proposed in He et al. (2016) in Section G of the appendix.*

### 3.3 BLOCK-SPARSE FULLY-CONNECTED NEURAL NETWORKS

In this section, we mathematically define FNNs we consider in this paper, in parallel with the CNN case. Our FNN, which we coin a *block-sparse* FNN, consists of $M$ possibly dense FNNs (blocks) concatenated in parallel, followed by a single fully-connected layer. We sketch the architecture of a block-sparse FNN in Figure 2.

**Definition 2** (Fully-connected Neural Networks (FNNs)). *Let $M \in \mathbb{N}_{>0}$ be the number of blocks in an FNN. Let $\boldsymbol{D}_m = (D_m^{(1)}, \ldots, D_m^{(L_m)}) \in \mathbb{N}_{>0}^{L_m}$ be the sequence of intermediate dimensions of the $m$-th block, where $L_m \in \mathbb{N}_{>0}$ is the depth of the $m$-th block for $m \in [M]$ [2]. Let $W_m^{(l)} \in \mathbb{R}^{D_m^{(l)} \times D_m^{(l-1)}}$ and $b_m^{(l)} \in \mathbb{R}$ be the weight matrix and the bias of the $l$-th layer of $m$-th block (with the convention $D_m^{(0)} = D$). Let $w_m \in \mathbb{R}^{D_m^{(L_m)}}$ be the weight (sub)vector of the final fully-connected layer corresponding to the $m$-th block and $b \in \mathbb{R}$ be the bias for the last layer. For $\boldsymbol{\theta} = ((W_m^{(l)})_{m,l}, (b_m^{(l)})_{m,l}, (w_m)_m, b)$ and an activation function $\sigma : \mathbb{R} \to \mathbb{R}$, we define $\mathrm{FNN}_{\boldsymbol{\theta}}^\sigma : \mathbb{R}^D \to \mathbb{R}$, the block-sparse FNN constructed from $\boldsymbol{\theta}$, by*

$$\mathrm{FNN}_{\boldsymbol{\theta}}^\sigma := \sum_{m=1}^M w_m^\top \mathrm{FC}_{\boldsymbol{W}_m, \boldsymbol{b}_m}^\sigma(\cdot) - b,$$

---

[2] Be aware that contrary to the CNN case, $m$ starts from 1 here.

*where* $\mathrm{FC}^{\sigma}_{\boldsymbol{W}_m, \boldsymbol{b}_m} := \mathrm{FC}^{\sigma}_{W_m^{(L_m)}, b_m^{(L_m)}} \circ \cdots \mathrm{FC}^{\sigma}_{W_m^{(1)}, b_m^{(1)}}.$

We call a block-sparse FNN with $M$ blocks a $M$-way block-sparse FNN. We say $\boldsymbol{\theta}$ is *compatible* with $(D_m^{(l)})_{m,l}$ when each component of $\boldsymbol{\theta}$ matches the dimension conditions determined by $(D_m^{(l)})_{m,l}$, as we did in the CNN case. Note that when $L_m = 1$ for all $m \in [M]$, the block-sparse FNN is a 2-layered neural network with $D' := \sum_{m=1}^{M} D_m^{(1)}$ hidden units of the form $f(x) = \sum_{d=1}^{D'} b_d \sigma(a_d^\top x - t_d) - b$ where $a_d \in \mathbb{R}^D$ and $b_d, t_d, b \in \mathbb{R}$.

For an architecture $\boldsymbol{D} = (D_m^{(l)})_{m \in [M], l \in [L_m]}$ and norm parameters for the block part $B^{(\mathrm{bs})} > 0$ and for the final layer $B^{(\mathrm{fin})} > 0$, we define $\mathcal{F}^{(\mathrm{FNN})} = \mathcal{F}^{(\mathrm{FNN})}_{\boldsymbol{D}, B^{(\mathrm{bs})}, B^{(\mathrm{fin})}}$, the set of function realizable by FNNs:

$$\mathcal{F}^{(\mathrm{FNN})} := \left\{ \mathrm{FNN}^{\mathrm{ReLU}}_{\boldsymbol{\theta}} \ \middle| \ \begin{array}{l} \boldsymbol{\theta} = ((W_m^{(l)})_{m,l}, (b_m^{(l)})_{m,l}, (w_m)_m, b) \text{ is compatible with } \boldsymbol{D}, \\ \max_{m \in [M], l \in [L_m]}(\|W_m^{(l)}\|_\infty \vee \|b_m^{(l)}\|_\infty) \le B^{(\mathrm{bs})}, \\ \max_{m \in [M]} \|w_m\|_\infty \vee |b| \le B^{(\mathrm{fin})}. \end{array} \right\}.$$

Again, the domain is restricted to $[-1, 1]^D$. Similar to the CNN case, we sometimes remove subscripts of the function class for simplicity.

## 4 MAIN THEOREMS

With the preparation in the previous sections, we state our main results of this paper. We only describe statements of theorems and corollaries and key ideas in the main article. All complete proofs are deferred to the appendix.

### 4.1 APPROXIMATION

Our first main theorem claims that any $M$-way block-sparse FNN is realizable by a ResNet-type CNN with fixed-sized channels and filters by adding $O(M)$ parameters, if we treat the widths $D_m^{(l)}$ of the FNN as constants with respect to $M$.

**Theorem 1.** *Let* $M \in \mathbb{N}_{>0}$, $K \in \{2, \ldots D\}$ *and* $L_0 := \left\lceil \frac{D-1}{K-1} \right\rceil$. *Let* $L_m \in \mathbb{N}_{>0}$, $D_m^{(l)} \in \mathbb{N}_{>0}$ ($m \in [M]$), *and* $\boldsymbol{D} = (D_m^{(l)})_{m \in [M], l \in [L_m]}$. *Then, there exist* $L'_m \in \mathbb{N}_{>0}$ ($m = 0, \ldots, M$), $\boldsymbol{C} = (C_m^{(l)})_{m=0,\ldots,M, l \in [L'_m]}$, *and* $\boldsymbol{K} = (K_m^{(l)})_{m=0,\ldots,M, l \in [L'_m]}$ *satisfying the following conditions:*

1. $L'_0 \le 1, L'_m \le L_m + L_0 \quad (m \in [M])$,

2. $\max\limits_{m=0,\ldots,M, l \in [L'_m]} C_m^{(l)} \le 4 \max\limits_{m \in [M], l \in [L_m]} D_m^{(l)}$, *and*

3. $\max\limits_{m=0,\ldots,M, l \in [L'_m]} K_m^{(l)} \le K$

*such that, for any* $B^{(\mathrm{bs})}, B^{(\mathrm{fin})} > 0$*, we have*

$$\mathcal{F}^{(\mathrm{FNN})}_{\boldsymbol{D}, B^{(\mathrm{bs})}, B^{(\mathrm{fin})}} \subset \mathcal{F}^{(\mathrm{CNN})}_{\boldsymbol{C}, \boldsymbol{K}, B^{(\mathrm{conv})}, B^{(\mathrm{fc})}}, \tag{1}$$

*that is, any FNN in* $\mathcal{F}^{(\mathrm{FNN})}_{\boldsymbol{D}, B^{(\mathrm{bs})}, B^{(\mathrm{fin})}}$ *can be realized by a CNN in* $\mathcal{F}^{(\mathrm{CNN})}_{\boldsymbol{C}, \boldsymbol{K}, B^{(\mathrm{conv})}, B^{(\mathrm{fc})}}$*. Here,* $B^{(\mathrm{conv})} = B^{(\mathrm{bs})}$ *and* $B^{(\mathrm{fc})} = B^{(\mathrm{fin})}\left(1 \vee \frac{1}{B^{(bs)}}\right)$*.*

An immediate consequence of this theorem is that if we can approximate a function $f^\circ$ with a block-sparse FNN, we can also approximate $f^\circ$ with a CNN.

### 4.2 ESTIMATION

Our second main theorem bounds the estimation error of the clipped ERM estimator $\hat{f}$.

**Theorem 2.** *Let $f^\circ : \mathbb{R}^D \to \mathbb{R}$ be a measurable function and $B^{(\mathrm{bs})}, B^{(\mathrm{fin})} > 0$. Let $M$, $K$, $L_0$, $L_m$, $\boldsymbol{D}$, $B^{(\mathrm{conv})}$ and $B^{(\mathrm{fc})}$ as in Theorem 1. Suppose $L_m', \boldsymbol{C}, \boldsymbol{K}$ satisfies the equation (1) of Theorem 1 for $B^{(\mathrm{bs})}$ and $B^{(\mathrm{fin})}$ (their existence is ensured for any $B^{(\mathrm{bs})}$ and $B^{(\mathrm{fin})}$ if they satisfy the conditions $1 - 3$. of Theorem 1). Suppose that the covering nubmer of $\mathcal{F}^{(\mathrm{CNN})} := \mathcal{F}_{\boldsymbol{C},\boldsymbol{K},B^{(\mathrm{conv})},B^{(\mathrm{fc})}}^{(\mathrm{CNN})}$ is larger than 3. Then, the clipped ERM estimator $\hat{f}$ in $\mathcal{F} := \{\mathrm{clip}[f] \mid f \in \mathcal{F}^{(\mathrm{CNN})}\}$ satisfies*

$$\mathbb{E}_{\mathcal{D}}\|\hat{f} - f^\circ\|_{\mathcal{L}^2(\mathcal{P}_X)}^2 \leq C \left( \inf_{f \in \mathcal{F}^{(\mathrm{FNN})}} \|f - f^\circ\|_\infty^2 + \frac{M_2 \tilde{F}^2}{N} \log(2M_1 BN) \right). \qquad (2)$$

*Here, $\mathcal{F}^{(\mathrm{FNN})} := \mathcal{F}_{\boldsymbol{D},B^{(\mathrm{bs})},B^{(\mathrm{fin})}}^{(\mathrm{FNN})}$, $C > 0$ is a universal constant, $\tilde{F} := \frac{\|f^\circ\|_\infty}{\sigma} \vee \frac{1}{2}$, and $B = B^{(\mathrm{conv})} \vee B^{(\mathrm{fc})}$. $M_1$ and $M_2$ are defined by*

$$M_1 := (2M + 3)C_0^{(L_0')} D(1 \vee B^{(\mathrm{fc})})(1 \vee B^{(\mathrm{conv})}) \left( \prod_{m=0}^M (1 + \rho_m) \right) \left( 1 + \sum_{m=0}^M L_m' \rho_m^+ \right),$$

$$M_2 := \sum_{m=0}^M \sum_{l=1}^{L_m'} \left( C_m^{(l-1)} C_m^{(l)} K_m^{(l)} + C_m^{(l)} \right) + C_0^{(L_0')} D + 1,$$

*where $\rho_m := \prod_{l=0}^{L_m'} C_m^{(l-1)} K_m^{(l)} B^{(\mathrm{conv})}$ and $\rho_m^+ := \prod_{l=0}^{L_m'} (1 \vee C_m^{(l-1)} K_m^{(l)} B^{(\mathrm{conv})})$.*

The first term of (2) is the approximation error achieved by $\mathcal{F}^{(\mathrm{FNN})}$. On the other hand, $M_1$ and $M_2$ are determined by the architectural parameters of $\mathcal{F}^{(\mathrm{CNN})}$ — $M_1$ corresponds to the Lipschitz constant of a function realized by a CNN and $M_2$ is the number of parameters, including zeros, of a CNN. Therefore, the second term of (2) represents the model complexity of $\mathcal{F}^{(\mathrm{CNN})}$. There is a trade-off between the two terms. Using appropriately chosen $M$ to balance them, we can evaluate the order of estimation error with respect to the sample size $N$.

**Corollary 1.** *Under the same assumptions as Theorem 2, suppose further $\log M_1(B^{(\mathrm{conv})} \vee B^{(\mathrm{fc})}) = \tilde{O}(1)$ as a function of $M$. If $\inf_{f \in \mathcal{F}^{(\mathrm{FNN})}} \|f - f^\circ\|_\infty^2 = \tilde{O}(M^{-\gamma_1})$ and $M_2 = \tilde{O}(M^{\gamma_2})$ for some constant $\gamma_1, \gamma_2 > 0$ independent of $M$, then, the clipped ERM estimator $\hat{f}$ of $\mathcal{F}$ achieves the estimation error $\|f^\circ - \hat{f}\|_{\mathcal{L}_2(\mathcal{P}_X)}^2 = \tilde{O}_p \left( N^{-\frac{2\gamma_1}{2\gamma_1 + \gamma_2}} \right)$.*

## 5 APPLICATION OF MAIN THEOREMS

### 5.1 BARRON CLASS

The Barron class is an example of the function class that can be approximated by block-sparse FNNs. We employ the definition of Barron functions used in Klusowski & Barron (2016).

**Definition 3** (Barron class). *We say a measurable function $f^\circ : [-1, 1]^D \to \mathbb{R}$ is a Barron function with the parameter $s > 0$ if $f^\circ$ admits the Fourier representation (i.e., $f^\circ(x) = \check{\mathcal{F}}\mathcal{F}[f^\circ]$) and $v_{f^\circ} := \int_{\mathbb{R}^D} \|w\|_2^s |\mathcal{F}[f^\circ](w)| \, \mathrm{d}w < \infty$. Here, $\mathcal{F}$ and $\check{\mathcal{F}}$ are the Fourier transformation and the inverse Fourier transformation, respectively.*

Klusowski & Barron (2016) studied the approximation of the Barron function $f^\circ$ with the parameter $s = 2$ by a linear combination of $M$ ridge functions (i.e., a 2-layered ReLU FNN). Specifically, they showed that there exists a function $f_M$ of the form

$$f_M := f^\circ(0) + \nabla f^{\circ\top}(0)x + \frac{1}{M} \sum_{m=1}^M b_m (a_m^\top x - t_m)_+ \qquad (3)$$

with $|b_m| \leq 1$, $\|a_m\|_1 = 1$ and $|t_m| \leq 1$, such that $\|f^\circ - f_M\|_\infty = \tilde{O}\left(M^{-\left(\frac{1}{2}+\frac{1}{D}\right)}\right)$. Using this approximator $f_M$, we can derive the same approximation order using CNNs by applying Theorem 1 with $L_1 = \cdots = L_M = 1$ and $D_1^{(1)} = \cdots = D_M^{(1)} = 1$.

**Corollary 2.** *Let $f^\circ : [-1,1]^D \to \mathbb{R}$ be a Barron function with the parameter $s = 2$ such that $f^\circ(0) = 0$ and $\nabla f^\circ(0) = \mathbf{0}_D$. Then, for any $K = 2, \ldots, D$, there exists a CNN $f^{(\mathrm{CNN})}$ with $M$ residual blocks, each of which has depth $O(1)$ and at most 4 channels, and whose filter size is at most $K$, such that $\|f^\circ - f^{(\mathrm{CNN})}\|_\infty = \tilde{O}\left(M^{-\left(\frac{1}{2} + \frac{1}{D}\right)}\right)$.*

We have one design choice when we apply Corollary 1 to derive the estimation error: how to set $B^{(\mathrm{bs})}$ and $B^{(\mathrm{fin})}$. Looking at (3), the naive choice would be $B^{(\mathrm{bs})} := 1$ and $B^{(\mathrm{fin})} := \frac{1}{M}$. However, this cannot satisfy the assumption on $M_1$ of Corollary 1, due to the term $\prod_{m=0}^{M}(1 + \rho_m)$ whose logarithm is $O(M)$. We want its logarithm to be $\tilde{O}(1)$. In order to do that, we change the *relative scale* between parameters in the block-sparse part and the fully-connected part using the homogeneous property of the ReLU function: $\mathrm{ReLU}(ax) = a\mathrm{ReLU}(x)$ for $a > 0$. The rescaling operation enables us to choose $B^{(\mathrm{bs})} := \frac{1}{M}$ and $B^{(\mathrm{fin})} = 1$ to meet the assumption of Corollary 1. By setting $\gamma_1 = \frac{1}{2} + \frac{1}{D}$ and $\gamma_2 = 1$, we obtain the desired estimation error.

**Corollary 3.** *There exist the number of residual blocks $M = O\left(N^{\frac{D}{2+2D}}\right)$, depth of each residual block $L = O(1)$, channel size $C = O(1)$, filter size $K \in \{2, \ldots, D\}$, and norm bounds for the convolution part $B^{(\mathrm{conv})} = O\left(N^{-\frac{D}{2+2D}}\right)$, and for the fully-connected part $B^{(\mathrm{fc})} = O\left(N^{\frac{D}{2+2D}}\right)$ such that for sufficiently large $N$, the clipped ERM estimator $\hat{f}$ of $\mathcal{F} := \{\mathrm{clip}[f] \mid f \in \mathcal{F}^{(\mathrm{CNN})}_{\mathbf{C},\mathbf{K},S,B^{(\mathrm{conv})},B^{(\mathrm{fc})}}\}$ achieves the estimation error $\|f^\circ - \hat{f}\|^2_{\mathcal{L}_2(\mathcal{P}_X)} = \tilde{O}_p\left(N^{-\frac{D+2}{2(D+1)}}\right)$. Here, $C_m^{(l)} = C, K_m^{(l)} = K$ for $m = 0, \ldots, M, l \in [L]$ and define $\mathbf{C} = (C_m^{(l)})_{m,l}, \mathbf{K} = (K_m^{(l)})_{m,l}$.*

### 5.2 Hölder Class

We next consider the approximation and error rates of CNNs when the true function is a $\beta$-Hölder function.

**Definition 4** (Hölder class). *Let $\beta > 0$, $f^\circ : [-1,1]^D \to \mathbb{R}$ is a $\beta$-Hölder function if*

$$\|f^\circ\|_\beta := \sum_{0 \le |\alpha| < \lfloor\beta\rfloor} \|\partial^\alpha f^\circ\|_\infty + \sum_{|\alpha| = \lfloor\beta\rfloor} \sup_{x \ne y} \frac{|\partial^\alpha f^\circ(x) - \partial^\alpha f^\circ(y)|}{|x - y|^{\beta - \lfloor\beta\rfloor}} < \infty.$$

*Here, $\alpha = (\alpha_1, \ldots, \alpha_D)$ is a multi-index. That is, $\partial^\alpha f := \frac{\partial^{|\alpha|} f}{\partial x_1^{\alpha_1} \ldots \partial x_D^{\alpha_D}}$ and $|\alpha| := \sum_{d=1}^D \alpha_d$.*

Yarotsky (2017) showed that FNNs with $O(S)$ non-zero parameters can approximate any $D$ variate $\beta$-Hölder function ($\beta > 0$) with the order of $\tilde{O}(S^{-\frac{\beta}{D}})$. Schmidt-Hieber (2017) also proved a similar statement using a different construction method. They only specified their width (Schmidt-Hieber (2017) only), depth, and non-zero parameter counts of the approximating FNN and did not write in detail how non-zero parameters are distributed explicitly in the statements (see Theorem 1 of Yarotsky (2017) and Theorem 5 of Schmidt-Hieber (2017)). However, if we carefully look at their proofs, we find that we can transform the FNNs they constructed into the block-sparse ones. Therefore, we can utilize these FNNs and apply Theorem 1. To meet the assumption of Corollary 1, we again rescale the parameters of the FNNs, as we did in the Barron class case, so that $\log M_1 = \tilde{O}(1)$. We can derive the approximation and estimation errors by setting $\gamma_1 = \frac{\beta}{D}$ and $\gamma_2 = 1$.

**Corollary 4.** *Let $\beta > 0$, and $f^\circ : [-1,1]^D \to \mathbb{R}$ be a $\beta$-Hölder function. Then, for any $K = 2, \ldots, D$, there exists a CNN $f^{(\mathrm{CNN})}$ with $O(M)$ residual blocks, each of which has depth $O(\log M)$ and $O(1)$ channels, and whose filter size is at most $K$, such that $\|f^\circ - f^{(\mathrm{CNN})}\|_\infty = \tilde{O}\left(M^{-\frac{\beta}{D}}\right)$.*

**Corollary 5.** *There exist the number of residual blocks $M = O\left(N^{\frac{D}{2\beta+D}}\right)$, depth of each residual block $L = \tilde{O}(1)$, channel size $C = O(1)$, filter size $K \in \{2, \ldots, D\}$, norm bounds for the convolution part $B^{(\mathrm{conv})} = O(1)$, and for the fully-connected part $B^{(\mathrm{fc})} > 0$ ($\log B^{(\mathrm{fc})} = O(\log N)$) such that for sufficiently large $N$, the clipped ERM estimator $\hat{f}$ of $\mathcal{F} := \{\mathrm{clip}[f] \mid f \in \mathcal{F}^{(\mathrm{CNN})}_{\mathbf{C},\mathbf{K},S,B^{(\mathrm{conv})},B^{(\mathrm{fc})}}\}$ achieves the estimation error $\|f^\circ - \hat{f}\|^2_{\mathcal{L}_2(\mathcal{P}_X)} = \tilde{O}_p\left(N^{-\frac{2\beta}{2\beta+D}}\right)$. Here, $C_m^{(l)} = C, K_m^{(l)} = K$ for $m = 0, \ldots, M, l \in [L]$ and define $\mathbf{C} = (C_m^{(l)})_{m,l}, \mathbf{K} = (K_m^{(l)})_{m,l}$.*

Since the estimation error rate of the $\beta$-Hölder class is $O_p\left(N^{-\frac{2\beta}{2\beta+D}}\right)$ (see, e.g., Tsybakov (2008)), Corollary 5 implies that our CNN can achieve the minimax optimal rate up to logarithmic factors even the width $D$, the channel size $C$, and the filter size $K$ are constant with respect to the sample size $N$.

## 6 CONCLUSION

In this paper, we established new approximation and statistical learning theories for CNNs by utilizing the ResNet-type architecture of CNNs and the block-sparse structure of FNNs. We proved that any $M$-way block-sparse FNN is realizable using CNNs with $O(M)$ additional parameters, when the width of the FNN is fixed. Using this result, we derived the approximation and estimation errors for CNNs from those for block-sparse FNNs. Our theory is general because it does not depend on a specific function class, as long as we can approximate it with block-sparse FNNs. To demonstrate the wide applicability of our results, we derived the approximation and error rates for the Barron class and Hölder class in almost same manner and showed that the estimation error of CNNs is same as that of FNNs, even if the CNNs have a constant channel size, filter size, and width with respect to the sample size. The key techniques were careful evaluations of the Lipschitz constant of CNNs and non-trivial weight parameter rescaling of FNNs.

One of the interesting open questions is the role of the weight rescaling. We critically use the homogeneous property of the ReLU activation function to change the relative scale between the block-sparse part and the fully-connected part, if it were not for this property, the estimation error rate would be worse. The general theory for rescaling, not restricted to the Barron nor Hölder class would be beneficial for deeper understanding of the relationship between the approximation and estimation capabilities of FNNs and CNNs.

Another question is when the approximation and estimation error rates of CNNs can *exceed* that of FNNs. We can derive the same rates as FNNs essentially because we can realize block-sparse FNNs using CNNs that have the same order of parameters (see Theorem 1). Therefore, if we dig into the internal structure of FNNs, like repetition, more carefully, the CNNs might need fewer parameters and can achieve better estimation error rate. Note that there is no hope to enhance this rate for the Hölder case (up to logarithmic factors) because the estimation rate using FNNs is already minimax optimal. It is left for future research which function classes and constraints of FNNs, like block-sparseness, we should choose.

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

## A  NOTATION

For tensor $a$, $a_+ := a \vee 0$ where maximum operation is performed in element-wise manner. Similarly $a_- := -(-a \vee 0)$. Note that $a = a_+ - a_-$ holds for any tensor $a$. For normed spaces $(V, \| \cdot \|_V), (W, \| \cdot \|_W)$ and linear operator $T : V \rightarrow W$ we denote the operator norm of $T$ by $\|T\|_{\mathrm{op}} := \sup_{\|v\|_V=1} \|Tv\|_W$. For a sequence $\boldsymbol{w} = (w^{(1)}, \ldots, w^{(L)})$ and $l \leq l'$, we denote its subsequence from the $l$-th to $l'$-th elements by $\boldsymbol{w}[l : l'] := (w^{(l)}, \ldots, w^{(l')})$. $\mathbf{1}_P$ equals to 1 if the statement $P$ is true, equals to 0 otherwise.

## B  PROOF OVERVIEW

### B.1  THEOREM 1

For $f^{(\mathrm{FNN})} \in \mathcal{F}^{(\mathrm{FNN})}$, we realize a CNN $f^{(\mathrm{CNN})}$ using $M$ residual blocks by "serializing" blocks in the FNN and converting them into convolution layers.

First, we double the channel size using the $m = 0$ part of CNN (i.e., $D_0^{(L_0)} = 2$). We will use the first channel for storing the original input signal for feeding to downstream (i.e., $m \geq 1$) blocks and the second one for accumulating the output of each blocks, that is, $\sum_{m=1}^{m'} w_m^\top \mathrm{FC}_{\boldsymbol{W}_m, \boldsymbol{b}_m}^{\mathrm{ReLU}}(x)$ where $w_m$ is the weight of the final fully-connected layer corresponding to the $m$-th dense block.

For $m = 1, \ldots, M$, we create the $m$-th residual block from the $m$-th block of $f^{(\mathrm{FNN})}$. First, we show that for any $a \in \mathbb{R}^D$ and $t \in \mathbb{R}$, there exists $L_0$-layered 4-channel ReLU CNN with $O(D)$ parameters whose first output coordinate equals to a ridge function $x \mapsto (a^\top x - t)_+$ (Lemma 1 and Lemma 2). Since the first layer of $m$-th block is concatenation of $D_m^{(1)}$ hinge functions, it is realizable by a $4D_m^{(1)}$-channel ReLU CNN with $L_0$-layers.

For the $l$-th layer of the $m$-th block ($m \in [M], l = 2, \ldots, L_m^{(l)}$), we prepare $D_m^{(l)}$ size-1 filters made from the weight parameters of the corresponding layer of the FNN. Observing that the convolution operation with size-1 filter is equivalent to a dimension-wise affine transformation, the first coordinate of the output of $l$-th layer of the CNN is inductively same as that of the $m$-th block of the FNN. After computing the $m$-th block FNN using convolutions, we add its output to the accumulating channel in the identity mapping.

Finally, we pick the first coordinate of the accumulating channel and subtract the bias term using the final affine transformation.

### B.2  THEOREM 2 AND COROLLARY 1

We relate the approximation error of Theorem 2 with the estimation error using the covering number of the hypothesis class $\mathcal{F}^{(\mathrm{CNN})}$. Although there are several theorems of this type, we employ the one in Schmidt-Hieber (2017) due to its convenient form (Lemma 5). We can prove that the logarithm of the covering number is upper bounded by $M_2 \log((B^{(\mathrm{conv})} \vee B^{(\mathrm{fc})})M_1/\varepsilon)$ (Lemma 4) using the similar techniques to the one in Schmidt-Hieber (2017). Theorem 2 is the immediate consequence of these two lemmas.

To prove Cororraly 1, we set $M = O(N^\alpha)$ for some $\alpha \geq 0$. Then, under the assumption of the corolarry, we have $\|f^\circ - \hat{f}\|_{\mathcal{L}^2(\mathcal{P}_x)}^2 = \tilde{O}\left(\max\left(N^{-2\alpha\gamma_1}, N^{\alpha\gamma_2-1}\right)\right)$ from Theorem 2. The order of the right hand side with respect to $N$ is minimized when $\alpha = \frac{1}{2\gamma_1+\gamma_2}$. By substituting $\alpha$, we can prove Corollary 1.

## C  PROOF OF THEOREM 1

### C.1  DECOMPOSITION OF AFFINE TRANSFORMATION

The following lemma shows that any affine transformation is realizable with a $\left\lceil \frac{D-1}{K-1} \right\rceil$-layered linear conventional CNN (without the final fully-connect layer).

**Lemma 1.** *Let $a \in \mathbb{R}^D$, $t \in \mathbb{R}$, $K \in \{2, \ldots, D-1\}$, and $L_0 := \left\lceil \frac{D-1}{K-1} \right\rceil$. Then, there exists*

$$w^{(l)} \in \begin{cases} \mathbb{R}^{K \times 2 \times 1} & \text{(for } l = 1) \\ \mathbb{R}^{K \times 2 \times 2} & \text{(for } l = 2, \ldots, L_0 - 1) \\ \mathbb{R}^{K \times 1 \times 2} & \text{(for } l = L_0) \end{cases}$$

*and $b \in \mathbb{R}$ such that*

1. $\displaystyle\sum_{l=1]}^{L_0} \|w^{(l)}\|_0 + \sum_{l=1}^{L_0} \|b^{(l)}\|_0 \le D + L_0,$

2. $\displaystyle\max_{l \in [L_o]} \|w_m\|_\infty = \|a\|_\infty, \ \max_{l \in [L_0]} \|b^{(l)}\|_\infty = |t|,$ *and*

3. $\mathrm{Conv}_{\boldsymbol{w}, \boldsymbol{b}}^{\mathrm{id}} : \mathbb{R}^D \to \mathbb{R}^D$ *satisfies* $\mathrm{Conv}_{\boldsymbol{w}, \boldsymbol{b}}^{\mathrm{id}}(x) = a^\top x - t$ *for any $x \in [-1, 1]^D$.*

*Proof.* First, observe that the convolutional layer constructed from $u = \begin{bmatrix} u_1 & \cdots & u_K \end{bmatrix}^\top \in \mathbb{R}^{K \times 1 \times 1}$ takes the inner product with the first $K$ elements of the input signal: $L^u(x) = \sum_{k=1}^{K} u_k x_k$. In particular, $u = \begin{bmatrix} 0 & \cdots & 0 & 1 \end{bmatrix}^\top \in \mathbb{R}^{K \times 1 \times 1}$ works as the "left-translation" by $K - 1$. Therefore, we should define $\boldsymbol{w}$ so that it takes the inner product with the $K$ left-most elements in the first channel and shift the input signal by $K - 1$ with the second channel. Specifically, we define $\boldsymbol{w} = (w^{(1)}, \ldots, w^{(L_0)})$ by

$$(w^{(1)})_{:,1,:} = \begin{bmatrix} a_1 \\ \vdots \\ a_K \end{bmatrix}, \qquad (w^{(1)})_{:,2,:} = \begin{bmatrix} 0 \\ \vdots \\ 0 \\ 1 \end{bmatrix},$$

$$(w^{(l)})_{:,1,:} = \begin{bmatrix} 0 & a_{(l-1)K+1} \\ \vdots & \vdots \\ 0 & a_{lK} \end{bmatrix}, \quad (w^{(l)})_{:,2,:} = \begin{bmatrix} 0 & 0 \\ \vdots & \vdots \\ 0 & 0 \\ 1 & 0 \end{bmatrix},$$

$$(w^{(L_0)})_{:,1,:} = \begin{bmatrix} 0 & a_{(L_0-1)K+1} \\ \vdots & \vdots \\ 0 & a_D \\ 0 & 0 \\ \vdots & \vdots \\ 0 & 0 \end{bmatrix}.$$

We set $\boldsymbol{b} := (\underbrace{0, \ldots, 0}_{L_0 - 1 \text{ times}}, t)$. Then $\boldsymbol{w}$ and $\boldsymbol{b}$ satisfy the condition of the lemma. $\qquad\square$

## C.2 Transformation of a linear CNN into a ReLU CNN

The following lemma shows that we can convert any linear CNN to a ReLU CNN that has approximately 4 times larger parameters. This type of lemma is also found in Petersen & Voigtlaender (2017) (Lemma 2.3).

**Lemma 2.** *Let $C = (C^{(1)}, \ldots, C^{(L)}) \in \mathbb{N}_{>0}^L$ be channel sizes $\boldsymbol{K} = (K^{(1)}, \ldots, K^{(L)}) \in \mathbb{N}_{>0}^L$ be filter sizes. Let $w^{(l)} \in \mathbb{R}^{K^{(l)} \times C^l \times C^{(l)}}$ and $b^{(l)} \in \mathbb{R}^{(l)}$. Consider the linear convolution layers constructed from $\boldsymbol{w}$ and $\boldsymbol{b}$: $f_{\mathrm{id}} := \mathrm{Conv}_{\boldsymbol{w}, \boldsymbol{b}}^{\mathrm{id}} : \mathbb{R}^D \to \mathbb{R}^{D \times C^{(L)}} \mathbb{N}_{>0}^L$ where $\boldsymbol{w} = (w^{(l)})_l$ and $\boldsymbol{b} = (b^{(l)})_l$. Then, there exists a pair $\tilde{\boldsymbol{w}} = (\tilde{w}^{(l)})_{l \in [L]}, \tilde{\boldsymbol{b}} = (\tilde{b}^{(l)})_{l \in [L]}$ where $\tilde{w}^{(l)} \in \mathbb{R}^{K^{(l)} \times 2C^{(l)} \times 2C^{(l-1)}}$ and $\tilde{b}^{(l)} \in \mathbb{R}^{2C^{(l)}}$ such that*

1. $\displaystyle\sum_{l=1}^{L} \|\tilde{w}^{(l)}\|_0 \le 4 \sum_{l=1}^{L} \|w^{(l)}\|_0, \ \sum_{l=1}^{L} \|\tilde{b}^{(l)}\|_0 \le \sum_{l=1}^{L} \|b^{(l)}\|_0,$

2. $\max\limits_{l \in [L]} \|\tilde{w}^{(l)}\|_\infty = \max\limits_{l \in [L]} \|w^{(l)}\|_\infty$, $\max\limits_{l \in [L]} \|\tilde{b}^{(l)}\|_\infty = \max\limits_{l \in [L]} \|b^{(l)}\|_\infty$, and

3. $f_{\mathrm{ReLU}} := \mathrm{Conv}^{\mathrm{ReLU}}_{\tilde{w},\tilde{b}} : \mathbb{R}^D \to \mathbb{R}^{D \times 2C^{(L)}}$, satisfies $f_{\mathrm{ReLU}}(\cdot) = (f_{\mathrm{id}}(\cdot)_+, f_{\mathrm{id}}(\cdot)_-)$.

*Proof.* We define $\tilde{w}$ and $\tilde{b}$ as follows:

$$(\tilde{w}^{(1)})_{k,:,:} = \begin{bmatrix} (w^{(1)})_{k,:,:} \\ -(w^{(1)})_{k,:,:} \end{bmatrix} \text{ for } k = 1, \ldots, K^{(1)},$$

$$(\tilde{w}^{(l)})_{k,:,:} = \begin{bmatrix} (w^{(l)})_{k,:,:} & -(w^{(l)})_{k,:,:} \\ -(w^{(l)})_{k,:,:} & (w^{(l)})_{k,:,:} \end{bmatrix} \text{ for } k = 1, \cdots K^{(l)},$$

$$\tilde{b}^{(l)} = \begin{bmatrix} b^{(l)} \\ -b^{(l)} \end{bmatrix}$$

By definition, a pair $(\tilde{w}, \tilde{b})$ satisfies the conditions (1) and (2). For any $x \in \mathbb{R}^D$, we set $y^{(l)} := \mathrm{Conv}^{\mathrm{id}}_{w[1:l],b[1:l]}(x) \in \mathbb{R}^{C^{(l)} \times D}$. We will prove

$$\mathrm{Conv}^{\mathrm{ReLU}}_{\tilde{w}[1:l],\tilde{b}[1:l]}(x) = \begin{bmatrix} y^{(l)}_+ & y^{(l)}_- \end{bmatrix}^\top \tag{4}$$

for $l = 1, \ldots, L$ by induction. Note that we obtain $f_{\mathrm{ReLU}}(\cdot) = (f_{\mathrm{id}+}(\cdot), f_{\mathrm{id}-}(\cdot))$ by setting $l = L$. For $l = 1$, by definition of $\tilde{w}^{(1)}$ we have,

$$(\tilde{w}^{(1)})_{\alpha,:,:}x^{\beta,:} = \begin{bmatrix} (w^{(1)})_{\alpha,:,:}x^{\beta,:} \\ -(w^{(1)})_{\alpha,:,:}x^{\beta,:} \end{bmatrix}$$

for any $\alpha, \beta \in [D]$. Summing them up and using the definition of $\tilde{b}^{(1)}$ yield

$$[L^{\tilde{w}^{(1)}}(x) - \mathbf{1}_D \otimes \tilde{b}^{(1)}]^\top = \begin{bmatrix} L^{w^{(1)}}(x) - \mathbf{1}_D \otimes b^{(1)} \\ -\left(L^{w^{(1)}}(x) - \mathbf{1}_D \otimes b^{(1)}\right) \end{bmatrix}^\top$$

Suppose (4) holds up to $l$ ($l < L$), by the definition of $\tilde{w}^{(l+1)}$,

$$\begin{aligned}
(\tilde{w}^{(l+1)})_{\alpha,:,:} \begin{bmatrix} (y^{(l)}_+)^{\beta,:} \\ (y^{(l)}_-)^{\beta,:} \end{bmatrix} &= \begin{bmatrix} (w^{(l+1)})_{\alpha,:,:} & -(w^{(l+1)})_{\alpha,:,:} \\ -(w^{(l+1)})_{\alpha,:,:} & (w^{(l+1)})_{\alpha,:,:} \end{bmatrix} \begin{bmatrix} (y^{(l)}_+)^{\beta,:} \\ (y^{(l)}_-)^{\beta,:} \end{bmatrix} \\
&= \begin{bmatrix} (w^{(l+1)})_{\alpha,:,:}\left((y^{(l)}_+)^{\beta,:} - (y^{(l)}_-)^{\beta,:}\right) \\ -(w^{(l+1)})_{\alpha,:,:}\left((y^{(l)}_+)^{\beta,:} - (y^{(l)}_-)^{\beta,:}\right) \end{bmatrix} \\
&= \begin{bmatrix} (w^{(l+1)})_{\alpha,:,:}(y^{(l)})^{\beta,:} \\ -(w^{(l+1)})_{\alpha,:,:}(y^{(l)})^{\beta,:} \end{bmatrix}
\end{aligned}$$

for any $\alpha, \beta \in [D]$. Again, by taking the summation and using the definition of $\tilde{b}^{(l+1)}$, we get

$$[L^{\tilde{w}^{(l+1)}}([y^{(l)}_+, y^{(l)}_-]) - \mathbf{1}_D \otimes \tilde{b}^{(1)}]^\top = \begin{bmatrix} L^{w^{(l+1)}}(y^{(l)}) - \mathbf{1}_D \otimes b^{(l+1)} \\ -\left(L^{w^{(l+1)}}(y^{(l)}) - \mathbf{1}_D \otimes b^{(l+1)}\right) \end{bmatrix}^\top .$$

By applying ReLU, we get

$$\mathrm{Conv}^{p^{(l+1)},\mathrm{ReLU}}_{\tilde{w}^{(l+1)},\tilde{b}^{(l+1)}}\left([y^{(l)}_+, y^{(l)}_-]\right) = \mathrm{ReLU}\left([y^{(l+1)}, -y^{(l+1)}]\right). \tag{5}$$

By using the induction hypothesis, we get

$$\begin{aligned}
\mathrm{Conv}^{\mathrm{ReLU}}_{\tilde{w}[1:(l+1)],\tilde{b}[1:(l+1)]}(x) &= \mathrm{Conv}^{p^{(l+1)},\mathrm{ReLU}}_{\tilde{w}^{(l+1)},\tilde{b}^{(l+1)}}\left([y^{(l)}_+, y^{(l)}_-]\right) \\
&= \mathrm{ReLU}\left([y^{(l+1)}, -y^{(l+1)}]\right) \\
&= [y^{(l+1)}_+, -y^{(l+1)}_-]
\end{aligned}$$

Therefore, the claim holds for $l + 1$. By induction, the claim holds for $L$, which is what we want to prove. $\square$

### C.3 CONCATENATION OF CNNs

We can concatenate two CNNs with the same depths and filter sizes in parallel. Although it is almost trivial, we state it formally as a proposition. In the following proposition, $C^{(0)}$ and $C'^{(0)}$ is not necessarily 1.

**Proposition 1.** *Let* $\boldsymbol{C} = (C^{(l)})_{l \in [L]}$, $\boldsymbol{C}' = (C'^{(l)})_{l \in [L]}$, *and* $\boldsymbol{K} = (K^{(l)})_{l \in [L]} \in \mathbb{N}_{>0}^L$. *Let* $w^{(l)} \in \mathbb{R}^{K^{(l)} \times C^{(l)} \times C^{(l-1)}}$, $b \in \mathbb{R}^{C^{(l)}}$ *and denote* $\boldsymbol{w} = (w^{(l)})_l$ *and* $\boldsymbol{b} = (b^{(l)})_l$. *We define* $\boldsymbol{w}'$ *and* $\boldsymbol{b}'$ *in the same way, with the exception that* $C^{(l)}$ *is replaced with* $C'^{(l)}$. *We define* $\tilde{\boldsymbol{w}} = (\tilde{w}^{(1)}, \ldots, \tilde{w}^{(L)})$ *and* $\tilde{\boldsymbol{b}} = (\tilde{b}^{(1)}, \ldots, \tilde{b}^{(L)})$ *by*

$$(\tilde{w}^{(l)})_{k,:,:} := \begin{bmatrix} w^{(l)} & 0 \\ 0 & w'^{(l)} \end{bmatrix} \in \mathbb{R}^{(C^{(l)}+C'^{(l)}) \times (C^{(l-1)}+C'^{(l-1)})}$$

$$\tilde{b}^{(l)} := \begin{bmatrix} b^{(l)} \\ b'^{(l)} \end{bmatrix} \in \mathbb{R}^{(C^{(l)}+C'^{(l)})}$$

*for* $l \in [L]$ *and* $k \in [K^{(l)}]$. *Then, we have,*

$$\mathrm{Conv}^{\sigma}_{\tilde{\boldsymbol{w}}, \tilde{\boldsymbol{b}}}([x \quad x']) = \begin{bmatrix} \mathrm{Conv}^{\sigma}_{\boldsymbol{w}, \boldsymbol{b}}(x) & \mathrm{Conv}^{\sigma}_{\boldsymbol{w}', \boldsymbol{b}'}(x') \end{bmatrix}$$

*for any* $x, x' \in \mathbb{R}^{D \times C^{(0)}}$ *and any* $\sigma : \mathbb{R} \to \mathbb{R}$. $\qquad\square$

Note that by the definition of $\| \cdot \|_0$ and $\| \cdot \|_\infty$, we have

$$\sum_{l=1}^{L} \|\tilde{w}^{(l)}\|_0 = \sum_{l=1}^{L} \|w^{(l)}\|_0 + \|w'^{(l)}\|_0,$$

$$\sum_{l=1}^{L} \|\tilde{b}^{(l)}\|_0 = \sum_{l=1}^{L} \|b^{(l)}\|_0 + \|b'^{(l)}\|_0,$$

$$\max_{l \in [L]} \|\tilde{w}^{(l)}\|_\infty = \max_{l \in [L]} \|w^{(l)}\|_\infty \vee \|w'^{(l)}\|_\infty, \quad \text{and}$$

$$\max_{l \in [L]} \|\tilde{b}^{(l)}\|_\infty = \max_{l \in [L]} \|b^{(l)}\|_\infty \vee \|b'^{(l)}\|_\infty.$$

### C.4 PROOF OF THEOREM 1

By the definition of $\mathcal{F}^{(\mathrm{FNN})}_{\boldsymbol{D}, B^{(\mathrm{bs})}, B^{(\mathrm{fin})}}$, there exists a 4-tuple $\boldsymbol{\theta} = ((W_m^{(l)})_{m,l}, (b_m^{(l)})_{m,l}, (w_m)_m, b)$ compatible with $(D_m^{(l)})_{m,l}$ ($m \in [M]$ and $l \in [L_m]$) such that

$$\max_{m \in [M], l \in [L_m]} (\|W_m^{(l)}\|_\infty \vee \|b_m^{(l)}\|_\infty) \le B^{(\mathrm{bs})}, \qquad \max_{m \in [M]} \|w_m\|_\infty \vee |b| \le B^{(\mathrm{fin})},$$

and $f^{(\mathrm{FNN})} = \mathrm{FNN}^{\mathrm{ReLU}}_{\boldsymbol{\theta}}$. We will construct the desired CNN consisting of $M$ residual blocks, whose $m$-th residual block is made from the ingredients of the corresponding $m$-th block in $f^{(\mathrm{FNN})}$ (specifically, $\boldsymbol{W}_m := (W_m^{(l)})_{l \in [L_m]}$, $\boldsymbol{b}_m := (b_m^{(l)})_{l \in [L_m]}$, and $w_m$).

**[The $m = 0$ Block]:** We prepare a single convolutional layer with 2 output channels and 2 size-1 filters suth that the first filter works as the identity function and the second filter inserts zeros to the second channel. Weight parameters of this convolutional layer are all zeros except single one. We denote this block by $\mathrm{Conv}_0$.

**[The $m = 1, \ldots, M$ Blocks]:** For fixed $m \in [M]$, we first create a CNN realizing $\mathrm{FC}^{\mathrm{ReLU}}_{\boldsymbol{W}_m, \boldsymbol{b}_m}$. We treat the first layer (i.e. $l = 1$) of $\mathrm{FC}^{\mathrm{ReLU}}_{\boldsymbol{W}_m, \boldsymbol{b}_m}$ as concatenation of $D_m^{(1)}$ hinge functions $\mathbb{R}^D \ni x \mapsto f_d(x) := ((W_m^{(1)})_d x - b_m^{(1)})_+$ for $d \in [D_m^{(1)}]$. Here, $(W_m^{(1)})_d \in \mathbb{R}^{1 \times D}$ is the $d$-th row of the matrix $W_m^{(1)} \in \mathbb{R}^{D_m^{(1)} \times D}$. We apply Lemma 1 and Lemma 2 and obtain ReLU CNNs realizing the hinge functions. By combining them in parallel using Proposition 1, we have a learnable parameter $\boldsymbol{\theta}_m^{(1)}$

such that the ReLU CNN $\text{Conv}^{\text{ReLU}}_{\boldsymbol{\theta}^{(1)}_m} : \mathbb{R}^{D \times 2} \to \mathbb{R}^{D \times 2D^{(1)}_m}$ constructed from $\boldsymbol{\theta}^{(1)}_m$ satisfies

$$\text{Conv}^{\text{ReLU}}_{\boldsymbol{\theta}^{(1)}_m}([x \quad x']^\top)_1 = \begin{bmatrix} f_1(x) & * & \cdots & f_{D^{(1)}_m}(x) & * \end{bmatrix}^\top.$$

Since we double the channel size in the $m = 0$ part, the identity mapping has 2 channels. Therefore, we made $\text{Conv}^{\text{ReLU}}_{\boldsymbol{\theta}^{(1)}_m}$ so that it has 2 input channels and neglects the input signals coming from the second one. This is possible by adding filters consisting of zeros appropriately.

Next, for $l$-th layer ($l = 2, \ldots, L_m$), we prepare size-1 filters $w^{(2)}_m \in \mathbb{R}^{1 \times D^{(2)}_m \times 2D^{(1)}}_m$ for $l = 2$ and $w^{(l)}_m \in \mathbb{R}^{1 \times D^{(l)}_m \times 2D^{(l-1)}_m}$ for $l = 3, \ldots, D^{(L_m)}_m$ defined by

$$(w^{(l)}_m)_{1,:,:} := \begin{cases} W^{(2)}_m \otimes \begin{bmatrix} 1 & 0 \end{bmatrix} & \text{if } l = 2 \\ W^{(l)}_m & \text{if } l = 3, \ldots, D^{(L_m)}_m, \end{cases}$$

where $\otimes$ is the Kronecker product of matrices. Intuitively, the $l = 2$ layer will pick all odd indices of the output of $\text{Conv}^{\text{ReLU}}_{\boldsymbol{\theta}^{(1)}_m}$ and apply the fully-connected layer. Note that $\text{Conv}^{\text{ReLU}}_{\theta^{(l)}_m}$ ($l \geq 2$) just rearranges parameters of $\text{FC}^{\text{ReLU}}_{\boldsymbol{W}_m, \boldsymbol{b}_m}$.

We construct CNNs from $\theta^{(l)}_m := (w^{(l)}_m, b^{(l)}_m)$ ($l \geq 2$) and concatenate them along with $\text{Conv}^{\text{ReLU}}_{\boldsymbol{\theta}^{(1)}_m}$:

$$\text{Conv}_m := \text{Conv}^{\text{ReLU}}_{\theta^{(L_m)}_m} \circ \cdots \circ \text{Conv}^{\text{ReLU}}_{\theta^{(2)}_m} \circ \text{Conv}^{\text{ReLU}}_{\boldsymbol{\theta}^{(1)}_m}.$$

The output dimension of $\text{Conv}_m$ is either $\mathbb{R}^{D \times 2D^{(L_m)}_m}$ (if $L_m = 1$) or $\mathbb{R}^{D \times D^{(L_m)}_m}$ (if $L_m \geq 2$)., We denote the output channel size (either $2D^{(L_m)}_m$ or $D^{(L_m)}_m$) by $D^{(\text{out})}_m$. By the inductive calculation, we have

$$\text{Conv}_m(x)_1 = \begin{cases} \text{FC}^{\text{ReLU}}_{\boldsymbol{W}_m, \boldsymbol{b}_m}(x) \otimes \begin{bmatrix} 1 & 0 \end{bmatrix} & \text{if } L_m = 1 \\ \text{FC}^{\text{ReLU}}_{\boldsymbol{W}_m, \boldsymbol{b}_m}(x) & \text{if } L_m \geq 2 \end{cases}.$$

By definition, $\text{Conv}_m$ has the depth of $L_0 + L_m - 1$, at most $4D^{(1)}_m \vee \max_{l=2,\ldots L_m} D^{(l)}_m \leq 4 \max_{l \in [L_m]} D^{(l)}_m$ channels. The $\infty$-norm of its parameters does not exceed that of parameters in $\text{FC}^{\text{ReLU}}_{\boldsymbol{W}_m, \boldsymbol{b}_m}$.

Next, we consider the filter $\tilde{w}_m \in \mathbb{R}^{1 \times 2 \times D^{(\text{out})}_m}$ defined by

$$(\tilde{w}_m)_{1,:,:} = \frac{B^{(\text{bs})}}{B^{(\text{fin})}} \begin{cases} \begin{bmatrix} 0 & \cdots & 0 \\ w_m \otimes \begin{bmatrix} 0 & 1 \end{bmatrix} \end{bmatrix} & \text{if } L_m = 1 \\ \begin{bmatrix} 0 & \cdots & 0 \\ & w_m \end{bmatrix} & \text{if } L_m \geq 2 \end{cases},$$

Then, $\text{Conv}'_m := \text{Conv}^{\text{id}}_{\tilde{w}_m, 0}$ adds the output of $m$-th residual block, weighted by $w_m$, to the second channel in the identity connections, while keeping the first channel intact. Note that the final layer of each residual block does not have the ReLU activation. By definition, $\text{Conv}'_m$ has $D^{(L_m)}_m$ parameters.

Given $\text{Conv}_m$ and $\text{Conv}'_m$ for each $m \in [M]$, we construct a CNN realizing $\text{FNN}^{\text{ReLU}}_{\boldsymbol{\theta}}$. Let $f^{(\text{conv})} : \mathbb{R}^D \to \mathbb{R}^D$ be the sequential interleaving concatenation of $\text{Conv}_m$ and $\text{Conv}'_m$, that is,

$$f^{(\text{conv})} := (\text{Conv}'_M \circ \text{Conv}_M + I) \circ \cdots \circ (\text{Conv}'_1 \circ \text{Conv}_1 + I) \circ \text{Conv}_0.$$

Then, we have

$$f^{(\text{conv})}_1 = \frac{B^{(\text{bs})}}{B^{(\text{fin})}} \sum_{m=1}^{M} w^\top_m \text{FC}^{\text{ReLU}}_{\boldsymbol{W}_m, \boldsymbol{b}_m}$$

(the subscript 1 represents the first coordinate).

**[Final Fully-connected Layer]** Finally, we set $w := \begin{bmatrix} \frac{B^{(\text{fin})}}{B^{(\text{bs})}} & 0 & \cdots & 0 \end{bmatrix} \in \mathbb{R}^D$ and put $\text{FC}^{\text{id}}_{w,b}$ on top of $f^{(\text{conv})}$ to pick the first coordinate of $f^{(\text{conv})}$ and subtract the bias term. By definition, $f^{(\text{CNN})} := \text{FC}^{\text{id}}_{w,b} \circ f^{(\text{conv})}$ satisfies $f^{(\text{CNN})} = f^{(\text{FNN})}$.

[**Condition Check**]: We will check $f^{(\mathrm{FNN})}$ satisfies the desired conditions. **(Condition 1)**: By definition the 0-th residual block $\mathrm{Conv}_0$ has $L_0' = 1$ layer. Since $\mathrm{Conv}_m$ and $\mathrm{Conv}_m'$ has $L_0 + L_m - 1$ and 1 layers, respectively, the $m(\geq 1)$-th residual block of $f^{(\mathrm{CNN})}$ has $L_m' = L_0 + L_m$ layers. **(Condition 2)**: $\mathrm{Conv}_m$ has at most $4 \max_{l \in [L_m]} D_m^{(l)}$ channels and $\mathrm{Conv}_m'$ has at most 2 channels, respectively. Therefore, the channel size of $f^{(\mathrm{CNN})}$ is at most $4 \max_{m \in [M], l \in [L_m]} D_m^{(l)}$. **(Condition 3)**: Since each filter of $\mathrm{Conv}^{(m)}$ and $\mathrm{Conv}_m'$ is at most $K$, the filter size of CNN is also at most $K$. **(Conditions on $B^{(\mathrm{conv})}$ and $B^{(\mathrm{fin})}$)**: Parameters of $f^{(\mathrm{conv})}$ are either 0, or parameters of $\mathrm{FC}_{\boldsymbol{W}_m, \boldsymbol{W}_m}^{\mathrm{ReLU}}$, whose absolute value is bounded by $B^{(\mathrm{bs})}$, or $\frac{B^{(\mathrm{bs})}}{B^{(\mathrm{fin})}} w_m$. Since we have $\|w_m\|_\infty \leq B^{(\mathrm{fin})}$, the $\infty$-norm of parameters in $f^{(\mathrm{CNN})}$ is bounded by $B^{(\mathrm{bs})}$. The parameters of the final fully-connected layer $\mathrm{FC}_{w, b}$ is either $B^{(\mathrm{fin})}$, 0, or $b$, therefore their norm is bounded by $\frac{B^{(\mathrm{fin})}}{B^{(\mathrm{bs})}} \vee B^{(\mathrm{fin})}$. □

**Remark 3.** *Another way to construct a CNN which is identical (as a function) to a given FNN is as follows. First, we use a "rotation" convolution with $D$ filters, each of which has a size $D$, to serialize all input signals to channels of a single input dimension. Then, apply size-1 convolution layers, whose $l$-th layer consisting of appropriately arranged weight parameters of the $l$-th layer of the FNN. This is essentially what Petersen & Voigtlaender (2018) does to prove the existence of a CNN equivalent to a given FNN. To restrict the size of filters to $K$, we should further replace the the first convolution layer with $O(D/K)$ convolution layers with size-$K$ filters. We can show essentially same statement using this construction method.*

# D   PROOF OF THEOREM 2

## D.1   COVERING NUMBER OF CNNS

The goal of this section is to prove Lemma 4, stated in Section D.1.5, that evaluates the covering number of the set of functions realized by CNNs $\mathcal{F}^{(\mathrm{CNN})}$.

### D.1.1   BOUNDS FOR CONVOLUTIONAL LAYERS

We assume $w, w' \in \mathbb{R}^{K \times J \times I}$, $b, b' \in \mathbb{R}$, and $x \in \mathbb{R}^{D \times I}$ unless specified. We have in mind that the activation function $\sigma$ is either the ReLU function or the identity function id. But the following proposition holds for any 1-Lipschitz function such that $\sigma(0) = 0$. Remember that we can treat $L^w$ as a linear operator from $\mathbb{R}^{D \times I}$ to $\mathbb{R}^{D \times J}$. We endow $\mathbb{R}^{D \times I}$ and $\mathbb{R}^{D \times J}$ with the sup norm and denote the operator norm $L^w$ by $\|L^w\|_{\mathrm{op}}$.

**Proposition 2.** *It holds that $\|L^w\|_{\mathrm{op}} \leq IK\|w\|_\infty$.*

*Proof.* Write $w = (w_{kji})_{k \in [K], j \in [J], i \in [I]}$, $L^w = ((L^w)_{\alpha, i}^{\beta, j})_{\alpha, \beta \in [D], j \in [J], i \in [I]}$. For any $x = (x^{\alpha, i})_{\alpha \in [D], i \in [I]} \in \mathbb{R}^{D \times I}$, the sup norm of $y := (y^{\beta j})_{\beta \in [D] j \in [J]} = L^w(x)$ is evaluated as follows:

$$
\begin{aligned}
\|y\|_\infty &= \max_{\beta, j} |y^{\beta, j}| \\
&\leq \max_{\beta, j} \sum_{\alpha, i} |(L^w)_{\alpha, i}^{\beta, j}| |x^{\alpha, i}| \\
&\leq \max_{\beta, j} \sum_{\alpha, i} |(L^w)_{\alpha, i}^{\beta, j}| \|x\|_\infty \\
&= \max_{\beta, j} \sum_{\alpha, i} |w_{(\alpha - \beta + 1), j, i}| \|x\|_\infty \\
&\leq \max_{\beta, j} \sum_{\alpha, i} \left( \mathbf{1}\{w_{(\alpha - \beta + 1), j, i} \neq 0\} \right) \|w\|_\infty \|x\|_\infty \\
&\leq IK\|w\|_\infty \|x\|_\infty
\end{aligned}
$$

□

**Proposition 3.** *It holds that* $\|\text{Conv}^\sigma_{w,b}(x)\|_\infty \leq \|L^w\|_{\text{op}}\|x\|_\infty + |b|.$

*Proof.*

$$
\begin{aligned}
\|\text{Conv}^\sigma_{w,b}(x)\|_\infty &\leq \|\sigma(L^w(x) - \mathbf{1}_D \otimes b)\|_\infty \\
&\leq \|L^w(x) - \mathbf{1}_D \otimes b\|_\infty \\
&\leq \|L^w(x)\|_\infty + \|\mathbf{1}_D \otimes b\|_\infty \\
&\leq \|L^w\|_{\text{op}}\|x\|_\infty + |b|.
\end{aligned}
$$

$\square$

**Proposition 4.** *The Lipschitz constant of* $\text{Conv}^\sigma_{w,b}$ *is bounded by* $\|L^w\|_{\text{op}}.$

*Proof.* For any $x, x' \in \mathbb{R}^{D \times I}$,

$$
\begin{aligned}
\|\text{Conv}^\sigma_{w,b}(x) - \text{Conv}^\sigma_{w,b}(x')\|_\infty &= \|\sigma\left(L^w(x) - \mathbf{1}_D \otimes b\right) - \sigma\left(L^w(x') - \mathbf{1}_D \otimes b\right)\|_\infty \\
&\leq \|\left(L^w(x) - \mathbf{1}_D \otimes b\right) - \left(L^w(x') - \mathbf{1}_D \otimes b\right)\|_\infty \\
&\leq \|L^w(x - x')\|_\infty \\
&\leq \|L^w\|_{\text{op}}\|x - x'\|_\infty.
\end{aligned}
$$

Note that the first inequality holds because the ReLU function is 1-Lipschitz. $\square$

**Proposition 5.** *It holds that* $\|\text{Conv}^\sigma_{w,b}(x) - \text{Conv}^\sigma_{w',b'}(x)\| \leq \|L^{w-w'}\|_{\text{op}}\|x\|_\infty + |b - b'|.$

*Proof.*

$$
\begin{aligned}
\|\text{Conv}^\sigma_{w,b}(x) - \text{Conv}^\sigma_{w',b'}(x)\| &= \|\sigma(L^w(x) - \mathbf{1}_D \otimes b) - \sigma(L^{w'}(x) - \mathbf{1}_D \otimes b')\|_\infty \\
&\leq \|(L^w(x) - \mathbf{1}_D \otimes b) - (L^{w'}(x) - \mathbf{1}_D \otimes b')\| \\
&= \|L^w(x) - L^{w'}(x)\| + \|\mathbf{1}_D \otimes (b - b')\|_\infty \\
&\leq \|L^{w-w'}\|_{\text{op}}\|x\|_\infty + |b - b'|
\end{aligned}
$$

$\square$

### D.1.2 BOUNDS FOR FULLY-CONNECTED LAYERS

In the following propositions in this subsection, we assume $W, W' \in \mathbb{R}^{D \times C}$, $b, b' \in \mathbb{R}$, and $x \in \mathbb{R}^{D \times C}$. Again, these propositions hold for any 1-Lipschitz function $\sigma : \mathbb{R} \to \mathbb{R}$ such that $\sigma(0) = 0$. But $\sigma = \text{ReLU}$ or id is enough for us.

**Proposition 6.** *It holds that* $|\text{FC}^\sigma_{W,b}(x)| \leq \|W\|_0 \|W\|_\infty \|x\|_\infty + |b|.$

*Proof.*

$$
\begin{aligned}
|\text{FC}^\sigma_{W,b}(x)| &\leq |\text{vec}(W)^\top \text{vec}(x) - b| \\
&\leq |\text{vec}(W)^\top \text{vec}(x)| + |b| \\
&\leq \sum_{\alpha,i} |W_{\alpha,i} x^{\alpha,i}| + |b|
\end{aligned}
$$

The number of non-zero summand in the summation is at most $\|W\|_0$ and each summand is bounded by $\|W\|_\infty \|x\|_\infty$ Therefore, we have $|\text{FC}^\sigma_{W,b}(x)| \leq \|W\|_0 \|W\|_\infty \|x\|_\infty + \|b\|_\infty.$ $\square$

**Proposition 7.** *The Lipschitz constant of* $\text{FC}^\sigma_{W,b}$ *is bounded by* $\|W\|_0 \|W\|_\infty.$

*Proof.* For any $x, x' \in \mathbb{R}^{D \times C}$,

$$
\begin{aligned}
|\text{FC}^\sigma_{W,b}(x) - \text{FC}^\sigma_{W,b}(x')| &\leq \|(\text{vec}(W)^\top \text{vec}(x) - b) - (\text{vec}(W)^\top \text{vec}(x') - b)\| \\
&\leq \|\text{vec}(W)^\top (\text{vec}(x) - \text{vec}(x'))\| \\
&\leq \|W\|_0 \|W\|_\infty \|\text{vec}(x) - \text{vec}(x')\|_\infty.
\end{aligned}
$$

$\square$

**Proposition 8.** *It holds that* $|\mathrm{FC}^\sigma_{W,b}(x) - \mathrm{FC}^\sigma_{W',b'}(x)| \leq (\|W\|_0 + \|W'\|_0)\|W - W'\|_\infty\|x\|_\infty + |b - b'|.$

*Proof.*

$$
\begin{aligned}
|\mathrm{FC}^\sigma_{W,b}(x) - \mathrm{FC}^\sigma_{W',b'}(x)| &\leq |(\mathrm{vec}(W)^\top \mathrm{vec}(x) - b) - (\mathrm{vec}(W')^\top \mathrm{vec}(x) - b')| \\
&= |(\mathrm{vec}(W - W')^\top \mathrm{vec}(x) - (b - b')| \\
&\leq |(\mathrm{vec}(W - W')^\top \mathrm{vec}(x)| + |b - b'| \\
&\leq \|W - W'\|_0\|W - W'\|_\infty\|x\|_\infty + |b - b'| \\
&\leq (\|W\|_0 + \|W'\|_0)\|W - W'\|_\infty\|x\|_\infty + |b - b'|
\end{aligned}
$$

$\square$

### D.1.3 Bounds for residual blocks

In this section, we denote the architecture of CNNs by $\boldsymbol{C} = (C^{(l)})_{l\in[L]} \in \mathbb{N}^L_{>0}$ and $\boldsymbol{K} = (K^{(l)})_{l\in[L]} \in \mathbb{N}^L_{>0}$ and the norm constraint on the convolution part by $B^{(\mathrm{conv})}$ ($C^{(0)}$ need not equal to 1 in this section). Let $w^l, w'^{(l)} \in \mathbb{R}^{K^{(l)}\times C^l \times C^{(l-1)}}$ and $b^{(l)}, b'^{(l)} \in \mathbb{R}$. We denote $\boldsymbol{w} := (w^{(l)})_{l\in[L]}, \boldsymbol{b} := (b^{(l)})_{l\in[L]}, \boldsymbol{w}' := (w'^{(l)})_{l\in[L]}$, and $\boldsymbol{b} := (b^{(l)})_{l\in[L]}$.

For $1 \leq l \leq l' \leq L$, we denote $\rho(l, l') := \prod_{i=l}^{l'}(C^{(i-1)}K^{(i)}B^{(\mathrm{conv})})$ and $\rho^+(l, l') := \prod_{i=l}^{l'} 1 \vee (C^{(i-1)}K^{(i)}B^{(\mathrm{conv})})$.

**Proposition 9.** *Let* $l \in [L]$. *We assume* $\max_{l\in[L]} \|w^{(l)}\|_\infty \vee \|b^{(l)}\|_\infty \leq B^{(\mathrm{conv})}$. *Then, for any* $x \in [-1, 1]^{D\times C^{(0)}}$, *we have* $\|\mathrm{Conv}^\sigma_{\boldsymbol{w}[1:l],\boldsymbol{b}[1:l]}(x)\|_\infty \leq \rho(1, l)\|x\|_\infty + B^{(\mathrm{conv})}l\rho^+(1, l).$

*Proof.* We write in shorthand as $C_{[s:t]} := \mathrm{Conv}^\sigma_{\boldsymbol{w}[s:t],\boldsymbol{b}[s:t]}$. Using Proposition 3 recursively, we get

$$
\|C_{[1:l]}(x)\|_\infty \leq \|L^{w^{(l)}}\|_{\mathrm{op}}\|C_{[1:l-1]}(x)\|_\infty + \|b^{(l)}\|_\infty
$$

$$\cdots$$

$$
\leq \|x\|_\infty \prod_{i=1}^l \|L^{w^{(i)}}\|_{\mathrm{op}} + \sum_{i=2}^l \|b^{(i-1)}\|_\infty \prod_{j=i}^l \|L^{w^{(j)}}\|_{\mathrm{op}} + \|b^{(l)}\|_\infty.
$$

By Proposition 2 and assumptions $\|w^{(i)}\|_\infty \leq B^{(\mathrm{conv})}$ and $\|b^{(i)}\|_\infty \leq B^{(\mathrm{conv})}$, it is further bounded by

$$
\|x\|_\infty \prod_{i=1}^l (C^{(i-1)}K^{(i)}B^{(\mathrm{conv})}) + B^{(\mathrm{conv})} \sum_{i=2}^l \prod_{j=i}^l (C^{(j-1)}K^{(j)}B^{(\mathrm{conv})}) + B^{(\mathrm{conv})}
$$

$$
\leq \rho(1, l)\|x\|_\infty + B^{(\mathrm{conv})}l\rho^+(1, l)
$$

$\square$

**Proposition 10.** *Let* $\varepsilon > 0$, *suppose* $\max_{l\in[L]} \|w^{(l)} - w'^{(l)}\|_\infty \leq \varepsilon$ *and* $\max_{l\in[L]} \|b^{(l)} - b'^{(l)}\|_\infty \leq \varepsilon$, *then* $\|C_{[1:L]} - C'_{[1:L]}(x)\|_\infty \leq (L\rho(1, L)\|x\|_\infty + (1 \vee B^{(\mathrm{conv})})L^2\rho^+(1, l))\varepsilon$ *for any* $x \in \mathbb{R}^{D\times C^{(0)}}$.

*Proof.* For any $l \in [L]$, we have

$$
\begin{aligned}
&\left|C'_{[l+1:L]} \circ (C_l - C'_l) \circ C_{[1:l-1]}(x)\right| \\
&\leq \|C'_{[l+1:L]} \circ (C_l - C'_l) \circ C_{[1:l-1]}(x)\|_\infty \\
&\leq \rho(l+1, L) \left\|(C_l - C'_l) \circ C_{[1:l-1]}(x)\right\|_\infty \quad \text{(by Proposition 2 and 4)} \\
&\leq \rho(l+1, L) \left(\rho(l, l)\|C_{[1:l-1]}\|_\infty\varepsilon + \varepsilon\right) \quad \text{(by Proposition 2 and 5)} \\
&\leq \rho(l+1, L) \left(\rho(l, l)(\rho(1, l-1)\|x\|_\infty + B^{(\mathrm{conv})}(l-1)\rho_+(1, l-1)) + 1\right)\varepsilon \quad \text{(by Proposition 9)}
\end{aligned}
$$

$$= \left( \rho(1, L) \|x\|_\infty + (1 \vee B^{(\text{conv})}) l \rho_+(1, L) \right) \varepsilon \tag{6}$$

Therefore,

$$\|C_{[1:L]}(x) - C'_{[1:L]}(x)\|_\infty \le \sum_{l=1}^{L} \|C_{[l+1:L]} \circ (C_l - C'_l) \circ C_{[1:l-1]}(x)\|_\infty$$

$$\le (L\rho(1, L) \|x\|_\infty + (1 \vee B^{(\text{conv})}) L^2 \rho^+(1, l)) \varepsilon$$

$\square$

### D.1.4 PUTTING THEM ALL

Let $M \in \mathbb{N}_{>0}$, $L_m \in \mathbb{N}_{>0}$, $C_m^{(l)}, K_m^{(l)} \in \mathbb{N}_{>0}$, $\boldsymbol{C} := (C_m^{(l)})_{m,l}$, and $\boldsymbol{K} := (K_m^{(l)})_{m,l}$ for $m = 0, \dots, M$ and $l \in [L_m]$. Let $\boldsymbol{\theta} = ((w_m^{(l)})_{m,l}, (b_m^{(l)})_{m,l}, W, b)$ and $\boldsymbol{\theta}' = ((w'_m^{(l)})_{m,l}, (b'_m^{(l)})_{m,l}, W', b')$ be tuples compatible with $(\boldsymbol{C}, \boldsymbol{K})$ such that $\text{CNN}_{\boldsymbol{\theta}}^{\text{ReLU}}, \text{CNN}_{\boldsymbol{\theta}'}^{\text{ReLU}} \in \mathcal{F}_{\boldsymbol{C}, \boldsymbol{K}, B^{(\text{conv})}, B^{(\text{fc})}}^{(\text{CNN})}$ for some $S \in \mathbb{N}_{>0}$ and $B^{(\text{conv})}, B^{(\text{fc})} > 0$. We denote the $l$-th convolution layer of the $m$-th block by $C_m^{(l)}$ and the $m$-th residual block of by $C_m$:

$$C_m^{(l)} := \begin{cases} \text{Conv}_{w_m^{(l)}}^{\text{id}} & (\text{if } l = L_m) \\ \text{Conv}_{w_m^{(l)}}^{\text{ReLU}} & (\text{otherwise}) \end{cases}$$

$$C_m := C_m^{(L_m)} \circ \cdots \circ C_m^{(1)}.$$

Also, we denote by $C_{[m:m']}$ the subnetwork of $\text{Conv}_{\boldsymbol{\theta}}^{\text{ReLU}}$ between the $m$-th and $m'$-th block. That is,

$$C_{[m:m']} := \begin{cases} (C_{m'} + I) \circ \cdots \circ (C_m + I) & (\text{if } m \ge 1) \\ (C_{m'} + I) \circ \cdots \circ C_m & (\text{if } m = 0) \end{cases}$$

for $m, m' = 0, \dots, M$. We define $C'^{(l)}_m$, $C'_m$ and $C'_{[m:m']}$ similarly for $\boldsymbol{\theta}'$.

**Proposition 11.** *For $m = 0, \dots M$ and $x \in [-1, 1]^D$, we have $\|C_{[0:m]}(x)\|_\infty \le (1 \vee B^{(\text{conv})}) \left( \prod_{i=0}^{m} (1 + \rho_i) \right) \left( 1 + \sum_{i=0}^{m} L_i \rho_i^+ \right)$. Here, $\rho_m$ and $\rho_m^+$ are constants defined in Theorem 2.*

*Proof.* By using Proposition 9 inductively, we have

$$\|C_{[0:m]}(x)\|_\infty \le \|C_m(C_{[0:m-1]}(x)) + C_{[0:m-1]}(x)\|_\infty$$

$$\le \|(1 + \rho_m)C_{[0:m-1]}(x) + B^{(\text{conv})} L_m \rho_{+m})\|_\infty$$

$$\le (1 + \rho_m)\|C_{[0:m-1]}(x)\|_\infty + B^{(\text{conv})} L_m \rho_m^+$$

$$\cdots$$

$$\le \|C_0(x)\|_\infty \prod_{i=1}^{m}(1 + \rho_i) + B^{(\text{conv})} \sum_{i=1}^{m} L_i \rho_i^+ \prod_{j=i+1}^{m}(1 + \rho_j)$$

$$\le \rho_0 \prod_{i=1}^{m}(1 + \rho_i) + B^{(\text{conv})} \sum_{i=0}^{m} L_i \rho_i^+ \prod_{j=i+1}^{m}(1 + \rho_j)$$

$$\le (1 \vee B^{(\text{conv})}) \left( \prod_{i=0}^{m}(1 + \rho_i) \right) \left( 1 + \sum_{i=0}^{m} L_i \rho_i^+ \right).$$

$\square$

**Lemma 3.** *Let $\varepsilon > 0$. Suppose $\boldsymbol{\theta}$ and $\boldsymbol{\theta}'$ are within distance $\varepsilon$, that is, $\max_{m,l} \|w_m^{(l)} - w'^{(l)}_m\|_\infty \le \varepsilon$, $\|b_m^{(l)} - b'^{(l)}_m\|_\infty \le \varepsilon$, $\|W - W'\|_\infty \le \varepsilon$, and $\|b - b'\|_\infty \le \varepsilon$. Then, $\|\text{CNN}_{\boldsymbol{\theta}}^{\text{ReLU}} - \text{CNN}_{\boldsymbol{\theta}'}^{\text{ReLU}}\|_\infty \le M_1 \varepsilon$ where $M_1$ is the function defined in Theorem 2.*

*Proof.* For any $x \in [-1, 1]^D$, we have

$$
\left| \text{CNN}_{\boldsymbol{\theta}}^{\text{ReLU}}(x) - \text{CNN}_{\boldsymbol{\theta}'}^{\text{ReLU}}(x) \right| = \left| \text{FC}_{W,b}^{\text{id}} \circ C_{[0:M]}(x) - \text{FC}_{W',b'}^{\text{id}} \circ C_{[0:M]}'(x) \right|
$$
$$
= \left| \left( \text{FC}_{W,b}^{\text{id}} - \text{FC}_{W',b'}^{\text{id}} \right) \circ C_{[0:M]}(x) \right|
$$
$$
+ \sum_{m=0}^{M} \left| \text{FC}_{W',b'}^{\text{id}} \circ C_{[m+1:M]} \circ (C_m - C_m') \circ C_{[0:m-1]}'(x) \right|. \quad (7)
$$

We will bound each term of (7). By Proposition 8 and Proposition 11,

$$
\left| \left( \text{FC}_{W,b}^{\text{id}} - \text{FC}_{W',b'}^{\text{id}} \right) \circ C_{[0:M]}(x) \right|
$$
$$
\leq (\|W\|_0 + \|W'\|_0)\|W - W'\|_\infty \|C_{[0:M]}(x)\|_\infty + \|b - b'\|_\infty
$$
$$
\leq 2C_0^{(L_0)} D \|C_{[0:M]}(x)\|_\infty \varepsilon + \varepsilon
$$
$$
\leq 2C_0^{(L_0)} D(1 \vee B^{\text{(conv)}}) \left( \prod_{m=0}^{M} (1 + \rho_m) \right) \left( 1 + \sum_{m=0}^{M} L_m \rho_m^+ \right) \varepsilon + \varepsilon
$$
$$
\leq 3C_0^{(L_0)} D(1 \vee B^{\text{(conv)}}) \left( \prod_{m=0}^{M} (1 + \rho_m) \right) \left( 1 + \sum_{m=0}^{M} L_m \rho_m^+ \right) \varepsilon \quad (8)
$$

On the other hand, for $m = 0, \ldots, M$,

$$
\left| \text{FC}_{W',b'}^{\text{id}} \circ C_{[m+1:M]}' \circ (C_m - C_m') \circ C_{[0:m-1]}(x) \right|
$$
$$
\leq \|W'\|_0 \|W'\|_\infty \|C_{[m+1:M]}' \circ (C_m - C_m') \circ C_{[1:m-1]}(x)\|_\infty \text{ (by Proposition 7)}
$$
$$
\leq C_0^{(L_0)} D B^{\text{(fc)}} \|C_{[m+1:M]}' \circ (C_m - C_m') \circ C_{[0:m-1]}(x)\|_\infty
$$
$$
\leq C_0^{(L_0)} D B^{\text{(fc)}} \left( \prod_{i=m+1}^{M} \rho_i \right) \|(C_m - C_m') \circ C_{[0:m-1]}(x)\|_\infty \text{ (by Proposition 2 and 4)}
$$
$$
\leq C_0^{(L_0)} D B^{\text{(fc)}} \left( \prod_{i=m+1}^{M} \rho_i \right) \left( \rho_m \|C_{[0:m-1]}\|_\infty \varepsilon + \varepsilon \right) \text{ (by Proposition 2 and 5)}
$$
$$
\leq C_0^{(L_0)} D B^{\text{(fc)}} \left( \prod_{i=m+1}^{M} \rho_i \right) \left( \rho_m (1 \vee B^{\text{(conv)}}) \left( \prod_{i=0}^{m-1} (1 + \rho_i) \right) \left( 1 + \sum_{i=0}^{m-1} L_i \rho_i^+ \right) + 1 \right) \varepsilon
$$
(by Proposition 9)
$$
\leq 2C_0^{(L_0)} D B^{\text{(fc)}} (1 \vee B^{\text{(conv)}}) \left( \prod_{i=0}^{M} (1 + \rho_i) \right) \left( 1 + \sum_{i=0}^{M} L_i \rho_i^+ \right) \varepsilon \quad (9)
$$

By applying (8) and (9) to (7), we have

$$
|\text{CNN}_{\boldsymbol{\theta}}^{\text{ReLU}}(x) - \text{CNN}_{\boldsymbol{\theta}'}^{\text{ReLU}}(x)|
$$
$$
\leq 3C_0^{(L_0)} D(1 \vee B^{\text{(conv)}}) \left( \prod_{m=0}^{M} (1 + \rho_m) \right) \left( 1 + \sum_{m=0}^{M} L_m \rho_m^+ \right) \varepsilon
$$
$$
+ 2M C_0^{(L_0)} D B^{\text{(fc)}} (1 \vee B^{\text{(conv)}}) \left( \prod_{m=0}^{M} (1 + \rho_m) \right) \left( 1 + \sum_{m=0}^{M} L_m \rho_m^+ \right) \varepsilon
$$
$$
\leq (2M + 3) C_0^{(L_0)} D(1 \vee B^{\text{(fc)}})(1 \vee B^{\text{(conv)}}) \left( \prod_{m=0}^{M} (1 + \rho_m) \right) \left( 1 + \sum_{m=0}^{M} L_m \rho_m^+ \right) \varepsilon
$$
$$
= M_1 \varepsilon.
$$

$\square$

### D.1.5 BOUNDS FOR COVERING NUMBER OF CNNS

For a metric space $(\mathcal{M}_0, d)$ and $\varepsilon > 0$, we denote the (external) covering number of $\mathcal{M} \subset \mathcal{M}_0$ by $\mathcal{N}(\varepsilon, \mathcal{M}, d)$: $\mathcal{N}(\varepsilon, \mathcal{M}, d) := \inf\{N \in \mathbb{N} \mid \exists f_1, \ldots, f_N \in \mathcal{M}_0 \text{ s.t. } \forall f \in \mathcal{M}, \exists n \in [N] \text{ s.t. } d(f, f_n) \leq \varepsilon\}$.

**Lemma 4.** *Let $B := B^{(\mathrm{conv})} \vee B^{(\mathrm{fc})}$. For $\varepsilon > 0$, we have $\mathcal{N}(\varepsilon, \mathcal{F}^{(\mathrm{CNN})}, \|\cdot\|_\infty) \leq \left(\frac{2BM_1}{\varepsilon}\right)^{M_2}$.*

*Proof.* The idea of the proof is same as that of Lemma 12 of Schmidt-Hieber (2017). We divide the interval of each parameter range ($[-B^{(\mathrm{conv})}, B^{(\mathrm{conv})}]$ or $[-B^{(\mathrm{fc})}, B^{(\mathrm{fc})}]$) into bins with width $\frac{\varepsilon}{M_1}$ (i.e., $2B^{(\mathrm{conv})}M_1\varepsilon^{-1}$ or $2B^{(\mathrm{fc})}M_1\varepsilon^{-1}$ bins for each interval). If $f, f' \in \mathcal{F}^{(\mathrm{CNN})}$ can be realized by parameters such that every pair of corresponding parameters are in a same bin, then, $\|f - f'\|_\infty \leq \varepsilon$ by Lemma 3. We make a subset $\mathcal{F}_0$ of $\mathcal{F}^{(\mathrm{CNN})}$ by picking up every combination of bins for $M_2$ parameters. Then, for each $f \in \mathcal{F}^{(\mathrm{CNN})}$, there exists $f_0 \in \mathcal{F}_0$ such that $\|f - f_0\|_\infty \leq \varepsilon$. There are at most $2BM_1\varepsilon^{-1}$ choices of bins for each parameter. Therefore, the cardinality of $\mathcal{F}_0$ is at most $\left(\frac{2BM_1}{\varepsilon}\right)^{M_2}$. $\qquad\square$

### D.2 PROOF OF THEOREM 2 AND COROLLARY 1

We use the lemma in Schmidt-Hieber (2017) to bound the estimation error of the clipped ERM estimator $\hat{f}$. Since our problem setting is slightly different from one in the paper, we restate the statement.

**Lemma 5** (cf. Schmidt-Hieber (2017) Lemma 10). *Let $\mathcal{F}$ be a family of measurable functions from $[-1, 1]^D$ to $\mathbb{R}$. Let $\hat{f}$ be the clipped ERM estimator of the regression problem described in Section 3.1. Suppose the covering number of $\mathcal{F}$ satisfies $\mathcal{N}(\varepsilon, \mathcal{F}, \|\cdot\|_\infty) \geq 3$. Then, $\mathbb{E}_{\mathcal{D}}\|f^\circ - \hat{f}\|^2_{\mathcal{L}^2(\mathcal{P}_X)} \leq 4\left(\inf_{f \in \mathcal{F}}\|f - f^\circ\|^2_{\mathcal{L}^2(\mathcal{P}_X)} + \left(56\log\mathcal{N}(\mathcal{F}, \frac{1}{N}, \|\cdot\|_\infty) + 180\right)\frac{\tilde{F}^2}{N}\right)$, where $\tilde{F} := \frac{R_{\mathcal{F}}}{\sigma} \vee \frac{\|f^\circ\|_\infty}{\sigma} \vee \frac{1}{2}$ and $R_{\mathcal{F}} := \sup\{\|f\|_\infty \mid f \in \mathcal{F}\}$.*

*Proof.* Basically, we convert our problem setting so that it fits to the assumptions of Lemma 10 of Schmidt-Hieber (2017) and apply the lemma to it. For $f : [-1, 1]^D \to [-\sigma\tilde{F}, \sigma\tilde{F}]$, we define $A[f] : [0, 1]^D \to [0, 2\tilde{F}]$ by $A[f](x') := \frac{1}{\sigma}f(2x'-1) + \tilde{F}$. Let $\hat{f}_1$ be the (non-clipped) ERM etimator of $\mathcal{F}$. We define $X' := \frac{1}{2}(X + 1)$, $f'^\circ := A[f^\circ]$, $Y' := f'^\circ(X) + \xi'$, $\mathcal{F}' := \{A[f] \mid f \in \mathcal{F}\}$, $\hat{f}'_1 := A[\hat{f}_1]$, and $\mathcal{D}' := ((x'_n, y'_n))_{n \in [N]}$ where $x'_n := \frac{1}{2}(x_n + 1)$ and $y'_n := f'^\circ(x'_n) + \frac{1}{\sigma}(y_n - f^\circ(x_n))$. Then, the probability that $\mathcal{D}'$ is drawn from $\mathcal{P}'^{\otimes N}$ is same as the probability that $\mathcal{D}$ is drawn from $\mathcal{P}^{\otimes N}$ where $\mathcal{P}'$ is the joint distribution of $(X', Y')$. Also, we can show that $\hat{f}'$ is the ERM estimator of the regression problem $Y' = f'^\circ + \xi'$ using the dataset $\mathcal{D}'$: $\hat{f}'_1 \in \arg\min_{f' \in \mathcal{F}'} \hat{\mathcal{R}}_{\mathcal{D}'}(f')$. We apply the Lemma 10 of Schmidt-Hieber (2017) with $n \leftarrow N$, $d \leftarrow D$, $\varepsilon \leftarrow 1$, $\delta \leftarrow \frac{1}{N}$, $\Delta_n \leftarrow 0$, $\mathcal{F}' \leftarrow \mathcal{F}$, $F \leftarrow 2\tilde{F}$, $\hat{f} \leftarrow \hat{f}'_1$ and use the fact that the estimation error of the clipped ERM estimator is no worse than that of the ERM estimator, that is, $\|f^\circ - \hat{f}\|^2_{\mathcal{L}^2(\mathcal{P}_X)} \leq \|f^\circ - \hat{f}_1\|^2_{\mathcal{L}^2(\mathcal{P}_X)}$ to conclude. $\quad\square$

*Proof of Theorem 2.* By definition of $\|\cdot\|_\infty$, we have $\|f - f^\circ\|_{\mathcal{L}^2(\mathcal{P}_X)} \leq \|f - f^\circ\|_\infty$ for any $f \in \mathcal{F}$. By Lemma 4, $\log\mathcal{N} := \log\mathcal{N}(\frac{1}{N}, \mathcal{F}^{(\mathrm{CNN})}, \|\cdot\|_\infty) \leq M_2\log(2BM_1N)$, where $B = B^{(\mathrm{conv})} \vee B^{(\mathrm{fc})}$. Therefore, by Lemma 5,

$$\|f^\circ - \hat{f}\|^2_{\mathcal{L}^2(\mathcal{P}_X)} \leq 4\left(\inf_{f \in \mathcal{F}}\|f - f^\circ\|^2_{\mathcal{L}^2(\mathcal{P}_X)} + (56\log\mathcal{N} + 180)\frac{\tilde{F}^2}{N}\right)$$

$$\leq C\left(\inf_{f \in \mathcal{F}}\|f - f^\circ\|^2_\infty + \frac{M_2\tilde{F}^2}{N}\log(2BM_1N)\right).$$

$\square$

*Proof of Corollay 1.* We only care the order with respect to $N$ in the $O$-notation. Set $M = \lfloor N^\alpha \rfloor$ for $\alpha \geq 0$. Using the assumptions of the corollary, the estimation error is

$$\|f^\circ - \hat{f}\|^2_{\mathcal{L}^2(\mathcal{P}_x)} = \tilde{O}\left(\max\left(N^{-2\alpha\gamma_1}, N^{\alpha\gamma_2 - 1}\right)\right).$$

by Theorem 2. The order of the right hand side with respect to $N$ is minimized when $\alpha = \frac{1}{2\gamma_1 + \gamma_2}$. By substituting $\alpha$, we can show Corollary 1. $\square$

## E   PROOF OF COROLLARY 2 AND COROLLARY 3

By Theorem 2 of Klusowski & Barron (2016), for each $M \in \mathbb{N}_{>0}$, there exists

$$f^{(\mathrm{FNN})} := \frac{1}{M}\sum_{m=1}^{M} b_m(a_m^\top x - t_m)_+ = \sum_{m=1}^{M} b_m \left(\frac{a_m^\top}{M}x - \frac{t_m}{M}\right)_+$$

with $|b_m| \leq 1$, $\|a_m\|_1 = 1$, and $|t_m| \leq 1$ such that $\|f^\circ - f^{(\mathrm{FNN})}\|_\infty \leq Cv_{f^\circ}\sqrt{\log M + D}M^{-\frac{1}{2}-\frac{1}{D}}$ where $C > 0$ is a universal constant. We set $L_m \leftarrow 1$, $D_m^{(1)} \leftarrow 1$, $B^{(\mathrm{bs})} \leftarrow \frac{1}{M}$, $B^{(\mathrm{fin})} \leftarrow 1$ ($m \in [M]$) in the Theorem 1, then, we have $f^{(\mathrm{FNN})} \in \mathcal{F}^{(\mathrm{FNN})}_{\boldsymbol{D}_1, B^{(\mathrm{bs})}, B^{(\mathrm{fin})}}$. By applying Theorem 1, there exists a CNN $f^{(\mathrm{CNN})} \in \mathcal{F}^{(\mathrm{CNN})}_{\boldsymbol{C}, \boldsymbol{K}, B^{(\mathrm{conv})}, B^{(\mathrm{fc})}}$ such that $f^{(\mathrm{FNN})} = f^{(\mathrm{CNN})}$. Here, $\boldsymbol{C} = (C_m^{(1)})_m$ with $C_m^{(1)} = 4$, $\boldsymbol{K} = (K_m^{(1)})_m$ with $K_m^{(1)} = K$, $B^{(\mathrm{conv})} = \frac{1}{M}$, and $B^{(\mathrm{fc})} = M$. This proves Corollary 2.

With these evaluations, we have $M_1 = O(M^3)$ (note that since $B^{(\mathrm{conv})} = \frac{1}{M}$, we have $\prod_{m=0}^{M}(1 + \rho_m) = O(1)$). In addition, $B^{(\mathrm{conv})}$ is $O(1)$ and $B^{(\mathrm{fc})}$ is $O(M)$. Therefore, we have $\log M_1 B = \tilde{O}(1)$. Since $M_2 = O(M)$, we can use Corollary 1 with $\gamma_1 = \frac{1}{2} + \frac{1}{D}$, $\gamma_2 = 1$. Since we have $M = O\left(N^{\frac{1}{2\gamma_1 + \gamma_2}}\right)$ by the proof of Corollary 1, we can derive the bounds for $B^{(\mathrm{conv})}$, and $B^{(\mathrm{fc})}$ with respect to $N$.

## F   PROOF OF COROLLARY 4 AND COROLLARY 5

We first prove the scaling property of the FNN class.

**Lemma 6.** *Let $M \in \mathbb{N}_{>0}$, $L_m \in \mathbb{N}_{>0}$, and $D_m^{(l)} \in \mathbb{N}_{>0}$ for $m \in [M]$ and $l \in [L_m]$. Let $B^{(\mathrm{bs})}, B^{(\mathrm{fin})} > 0$. Then, for any $k \geq 1$, we have $\mathcal{F}^{(\mathrm{FNN})}_{\boldsymbol{D}, B^{(\mathrm{bs})}, B^{(\mathrm{fin})}} \subset \mathcal{F}^{(\mathrm{FNN})}_{\boldsymbol{D}, k^{-1}B^{(\mathrm{bs})}, k^L B^{(\mathrm{fin})}}$ where $L := \max_{m \in [M]} L_m$ is the maximum depth of the blocks.*

*Proof.* Let $\boldsymbol{\theta} = ((W_m^{(l)})_{m,l}, (b_m^{(l)})_{m,l}, (w_m)_m, b)$ be the parameter of an FNN and suppose that $\mathrm{FNN}^{\mathrm{ReLU}}_{\boldsymbol{\theta}} \in \mathcal{F}^{(\mathrm{FNN})}_{\boldsymbol{D}, B^{(\mathrm{bs})}, B^{(\mathrm{fin})}}$. We define $\boldsymbol{\theta}' := ((W'^{(l)}_m)_{m,l}, (b'^{(l)}_m)_{m,l}, (w'_m), b')$ by

$$W'^{(l)}_m := k^{-\frac{L}{L_m}} W_m^{(l)} \qquad b'^{(l)}_m := k^{-l\frac{L}{L_m}} b_m^{(l)} \qquad w'_m := k^L w_m \qquad b' := b.$$

Since $k \geq 1$, we have $\mathrm{FNN}^{\mathrm{ReLU}}_{\boldsymbol{\theta}'} \in \mathcal{F}^{(\mathrm{FNN})}_{\boldsymbol{D}, k^{-1}B^{(\mathrm{bs})}, k^L B^{(\mathrm{fin})}}$. Also, by the homogeneous property of the ReLU function (i.e., $\mathrm{ReLU}(ax) = a\mathrm{ReLU}(x)$ for $a > 0$), we have $\mathrm{FNN}^{\mathrm{ReLU}}_{\boldsymbol{\theta}} = \mathrm{FNN}^{\mathrm{ReLU}}_{\boldsymbol{\theta}'}$. $\square$

Next, we prove the existence of a block-sparse FNN with constant-width blocks that optimally approximates a given $\beta$-Hölder function. It is almost same as the proof of Theorem 5 of Schmidt-Hieber (2017). However, we need to construct the FNN so that it has a block-sparse structure.

**Lemma 7** (cf. Schmidt-Hieber (2017) Theorem 5 ). *Let $\beta > 0$, $M \in \mathbb{N}_{>0}$ and $f^\circ : [-1, 1]^D \to \mathbb{R}$ be a $\beta$-Hölder function. Then, there exists $D' := O(1) \in \mathbb{N}_{>0}$, $L' := O(\log M) > 0$ ($C_1$ and $C_2$ are constants independent of $M$) and a block-sparse FNN $f^{(\mathrm{FNN})} \in \mathcal{F}^{(\mathrm{FNN})}_{\boldsymbol{D}, 1, 2M\|f^\circ\|_\beta}$ such that $\|f^\circ - f^{(\mathrm{FNN})}\|_\infty = \tilde{O}(M^{-\frac{\beta}{D}})$. Here, we set $L_m := L'$ and $D_m^{(l)} := D'$ for all $m \in [M]$ and $l \in [L_m]$ and define $\boldsymbol{D} := (D_m^{(l)})_{m,l}$.*

*Proof.* First, we prove the lemma when the domain of $f^\circ$ is $[0,1]^D$. Let $M'$ be the largest interger satisfying $(M'+1)^D \leq M$. Let $\Gamma(M') = \left(\frac{\mathbb{Z}}{M'}\right)^D \cap [0,1]^D = \{\frac{m'}{M'} \mid m' \in \{0,\ldots,M'\}^D\}$ be the set of lattice points in $[0,1]^{D3}$. Note that the cardinality of $\Gamma(M')$ is $(M'+1)^D$. Let $P_a^\beta f^\circ$ be the Taylor expansion of $f^\circ$ up to order $\lfloor\beta\rfloor$ at $a \in [0,1]^D$:

$$(P_a^\beta f^\circ)(x) = \sum_{0 \leq |\alpha| < \beta} \frac{(\partial^\alpha f^\circ)(a)}{\alpha!}(x-a)^\alpha.$$

For $a \in [0,1]^D$, we define a hat-shaped function $H_a : [0,1]^D \to [0,1]$ by

$$H_a(x) := \prod_{j=1}^D (M'^{-1} - |x_j - a_j|_+).$$

Note that we have $\sum_{a \in \Gamma(M')} H_a(x) = 1$, i.e., they are a partition of unity. Let $P^\beta f^\circ$ be the weighted sum of the Taylor expansions at lattice points of $\Gamma(M')$:

$$(P^\beta f^\circ)(x) := M'^D \sum_{a \in D(M')} (P_a^\beta f^\circ)(x) H_a(x).$$

By Lemma 7 of Schmidt-Hieber (2017), we have

$$\|P^\beta f^\circ - f^\circ\|_\infty \leq \|f^\circ\|_\beta M'^{-\beta}.$$

Let $m$ be an interger specified later and set $L^* := (m+5)\lceil\log_2 D\rceil$. By the proof of Lemma 8 of Schmidt-Hieber (2017), for any $a \in \Gamma(M')$, there exists an FNN $\text{Hat}_a : [0,1]^D \to [0,1]$ whose depth and width are at most $2 + L^*$ and $6D$, respectively and whose parameters have sup-norm 1, such that

$$\|\text{Hat}_a - H_a\|_\infty \leq 3^D 2^{-m}.$$

Next, let $B := 2\|f^\circ\|_\beta$ and $C_{D,\beta}$ be the number of distinct $D$-variate monomials of degree up to $\lfloor\beta\rfloor$. By the equation (7.11) of Schmidt-Hieber (2017), for any $a \in \Gamma(M)$, there exists an FNN $Q_a : [0,1]^D \to [0,1]$ [4] whose depth and width are $1 + L^*$ and $6DC_{D,\beta}$ respectively and whose parameters have sup-norm 1, such that

$$\left\|Q_a - \left(\frac{P_a^\beta f^\circ}{B} + \frac{1}{2}\right)\right\|_\infty \leq 3^D 2^{-m}.$$

Thirdly, by Lemma 4 of Schmidt-Hieber (2017), there exists an FNN $\text{Mult} : [0,1]^2 \to [0,1]$, whose depth and width are $m+4$ and 6, respectively and whose parameters have sup-norm 1 such that

$$|\text{Mult}(x,y) - xy| \leq 2^{-m}$$

for any $x, y \in [0,1]$. For each $a \in \Gamma(M')$, we combine $\text{Hat}_a$ and $Q_a$ using $\text{Mult}$ and constitute a block of the block-sparse FNN corresponding to $a \in \Gamma(M)$ by $\text{FC}_a := \text{Mult}(Q_a(\cdot), \text{Hat}_a(\cdot))$. Then, we have

$$\left\|\text{FC}_a - \left(\frac{P_a^\beta f^\circ}{B} + \frac{1}{2}\right)H_a\right\|_\infty \leq 2^{-m} + 3^D 2^{-m} + 3^D 2^{-m}$$

$$\leq 3^{D+1} 2^{-m}.$$

We define $f^{(\text{FNN})}(x) := \sum_{a \in \Gamma(M)}(BM'^D \text{FC}_a(x)) - \frac{B}{2}$. By construction, $f^{(\text{FNN})}$ is a block-sparse FNN with $(M'+1)^D(\leq M)$ blocks each of which has depth and width at most $L' := 2 + L^* + (m+4)$

---

[3]Schmidt-Hieber (2017) used $D(M')$ to denote this set of lattice points. We used different character to avoid notational conflict.

[4]We prepare $Q_a$ for each $a \in \Gamma(M)$ as opposed to the original proof of Schmidt-Hieber (2017), in which $Q_a$'s shared the layers the except the final one and were collectively denoted by $Q_1$.

and $D' := 6(C_{D,\beta} + 1)D$, respectively. The norms of the block-sparse part and the finally fully-connected layer are 1 and $BM'^D(\leq BM)$, respectively. In addition, we have

$$|f^{(\mathrm{FNN})}(x) - (P^\beta f^\circ)(x)|$$

$$\leq \sum_{a \in \Gamma(M)} BM'^D \left| \mathrm{FC}_a(x) - \left( \frac{(P_a^\beta f^\circ)(x)}{B} + \frac{1}{2} \right) H_a(x) \right| + \frac{B}{2} \left| 1 - M'^D \sum_{a \in \Gamma(M')} H_a(x) \right|$$

$$\leq (M' + 1)^D \times BM'^D 3^{D+1} 2^{-m}$$

$$\leq 3^{D+1} 2^{-m} BM^2$$

for any $x \in [0,1]^D$. Therefore,

$$|f^{(\mathrm{FNN})}(x) - f^\circ(x)| \leq |f^{(\mathrm{FNN})} - (P^\beta f^\circ)(x)| + |(P^\beta f^\circ)(x) - f^\circ(x)|$$

$$\leq B3^{D+1} M^2 2^{-m} + \|f^\circ\|_\beta M^{-\beta}$$

$$\leq 2\|f^\circ\|_\beta 3^{D+1} M^2 2^{-m} + \|f^\circ\|_\beta M^{-\frac{\beta}{D}} 2^\beta.$$

We set $m = \lceil \log_2 M^{2+\frac{\beta}{D}} \rceil$, then, we have $L' = O(\log M)$, $D' = O(1)$, and

$$\|f^{(\mathrm{FNN})} - f^\circ\| \leq \|f^\circ\|_\beta (2 \cdot 3^{D+1} + 2^\beta) M^{-\frac{\beta}{D}}.$$

By the defnition of $f^{(\mathrm{FNN})}$ we have $f^{(\mathrm{FNN})} \in \mathcal{F}_{\boldsymbol{D},1,2\|f^\circ\|_\beta M}^{(\mathrm{FNN})}$.

When the domain of $f^\circ$ is $[-1,1]^D$, we should add the function $x \mapsto \frac{1}{2}(x+1) = \frac{1}{2}(x+1)_+ - \frac{1}{2}(-x-1)_+$ as a first layer of each block to fit the range into $[0,1]^D$. Specifically, suppose the first layer of $m$-th block in $f^{(\mathrm{FNN})}$ is $x \mapsto \mathrm{ReLU}(Wx - b)$, then the first two layers become $x \mapsto \mathrm{ReLU}([\frac{1}{2}(x+1) \quad -\frac{1}{2}(x+1)])$ and $[y_1 \quad y_2] \mapsto \mathrm{ReLU}(Wy_1 - Wy_2 - b)$, respectively. Since this transformation does not change the maximum sup norm of parameters in the block-sparse and the order of $L'$ and $D'$, the resulting FNN is still belongs to $\mathcal{F}_{\boldsymbol{D},1,2\|f^\circ\|M}^{(\mathrm{FNN})}$. $\qquad\square$

*Proof of Corollary 4 and Corollary 5.* In this proof, we only care the dependence on $M$ in the $O$-notation. Let $\tilde{M} := 2\|f^\circ\|_\beta M$. By Lemma 7, there exists $f^{(\mathrm{FNN})} \in \mathcal{F}_{\boldsymbol{D},1,\tilde{M}}^{(\mathrm{FNN})}$ such that $\|f^{(\mathrm{FNN})} - f^\circ\|_\infty = O\left(M^{-\frac{\beta}{D}}\right)$ ($L'$, $D'$, and $\boldsymbol{D}$ as in Lemma 7). Let $k := 16D'K(M^{\frac{1}{L'}} \wedge 1)^{-1} = 16D'K(e^{\frac{1}{C'}} \wedge 1)^{-1} \geq 1$ where $C'$ is a constant such that $L' = C' \log M$. Using, Lemma 6, there exists $\tilde{f}^{(\mathrm{FNN})} \in \mathcal{F}_{\boldsymbol{D},k^{-1},k^{L'}\tilde{M}}^{(\mathrm{FNN})}$ such that $\tilde{f}^{(\mathrm{FNN})} = f^{(\mathrm{FNN})}$. We apply Theorem 1 to $\mathcal{F}_{\boldsymbol{D},k^{-1},k^{L'}\tilde{M}}^{(\mathrm{FNN})}$ and find $f^{(\mathrm{CNN})} \in \mathcal{F}_{\boldsymbol{C},\boldsymbol{K},B^{(\mathrm{conv})},B^{(\mathrm{fc})}}^{(\mathrm{CNN})}$ such that $L \leq M(L' + L_0)$, $\boldsymbol{C} := (C_m^{(l)})_{m=0,\ldots,M,l\in[L_m]}$ with $C_m^{(l)} \leq 4D'$, $\boldsymbol{K} := (K_m^{(l)})_{m=0,\ldots,M,l\in[L_m]}$ with $K_m^{(l)} \leq K$, $B^{(\mathrm{conv})} = k^{-1}$, $B^{(\mathrm{fc})} = k^{L'}(k \vee 1)\tilde{M} = k^{L'+1}\tilde{M}$, and $f^{(\mathrm{CNN})} = \tilde{f}^{(\mathrm{FNN})}$. By definition, we have $B^{(\mathrm{conv})} = k^{-1} = O(1)$ and $\log B^{(\mathrm{fc})} = (L' + 1)k + \log(\tilde{M}) = O(\log M)$. This proves Corollary 4.

By the definition of $k$ and the bound on $C_m^{(l)}$ and $K_m^{(l)}$, we have $C_m^{(l-1)} K_m^{(l)} k^{-1} \leq \frac{1}{4} M^{-\frac{1}{L'}}$. Therefore, we have $\rho_m \leq \prod_{l=1}^{L'}(C_m^{(l-1)} K_m^{(l)} k^{-1}) \leq M^{-1}$ and hence $\prod_{m=0}^{M}(1 + \rho_m) = O(1)$. Since $C_m^{(l-1)} K_m^{(l)} k^{-1} \leq \frac{1}{2}$ for sufficiently large $M$, we have $\rho_m^+ = 1$ for sufficiently large $M$. In addition, we have $\log(B^{(\mathrm{conv})} \vee B^{(\mathrm{fc})}) = \tilde{O}(1)$. Combining them, we have $\log M_1 = \tilde{O}(1)$ and hence $\log M_1(B^{(\mathrm{conv})} \vee B^{(\mathrm{fc})}) = \tilde{O}(1)$. For $M_2$, we can bound it by $M_2 = O(M \log M)$ using bounds for $C_m^{(l)}$, $K_m^{(l)}$ and $L'$. Therefore, we can apply Corollary 2 with $\gamma_1 = \frac{\beta}{D}$, $\gamma_2 = 1$ and obtain the desired estimation error. Since we have $M = O\left(N^{\frac{1}{2\gamma_1+\gamma_2}}\right)$ by the proof of Corollary 1, we can derive the bounds for $L_m$, $B^{(\mathrm{conv})}$, and $B^{(\mathrm{fc})}$ with respect to $N$. $\qquad\square$

# G    COMPARISON OF OUR CNNS AND ORIGINAL RESNET

There are several differences between the CNN in this paper and the original ResNet, aside from the number of layers. First and foremost, our CNN does not have pooling nor Batch Normalization

(Ioffe & Szegedy (2015)) layers. It is left for future research whether our result can extend to the ResNet-type CNNs with pooling or Batch Normalization layers. Second, our CNN does not have ReLU activation after the junction points and the final layer of the 0-th block, while they have in the original ResNet. We choose this design to make proofs simpler. We can easily extend our results to the architecture that adds the ReLU activations to those points with slight modifications using similar techniques appeared in Lemma 2 of the appendix.

