# OpenReview forum: "Approximation and non-parametric estimation of ResNet-type convolutional neural networks via block-sparse fully-connected neural networks"
_ICLR.cc/2019/Conference_

### Official Review · AnonReviewer1 · 2018-11-02
**Interesting approximation and estimation results, but considers somewhat unrealistic CNNs**

**Rating:** 4
**Confidence:** 3

**Review:**

The paper studies approximation and estimation properties of CNNs with residual blocks in the context
of non-parametric regression, by constructing equivalent fully-connected architectures (with a block-sparse structure),
and leveraging previous approximation results for such functions.
Explicit risk bounds are obtained for regression functions in Barron and Holder classes.

The main contribution of the paper is Theorem 1, which shows that a class of ResNet-type CNNs
contains a class of "block-sparse" fully-connected networks, with appropriate constraints on various size quantities.
This result allows the authors to obtain a general risk bound for the ResNet CNN that minimizes empirical risk
(Theorem 2, which mostly follows Schmidt-Hieber (2017)),
as well as adaptations of the bound for the Barron and Holder classes, by relying on existing approximation results.

The construction of Theorem 1 is interesting, and shows that ResNet CNNs can be quite powerful function approximators,
even with a filter size that is arbitrarily fixed.
However, the obtained CNN approximating architectures look quite unrealistic compared to most practical use-cases of CNNs,
since they specifically try to reproduce a fully-connected architecture, leading to residual blocks of depth ~= D/K,
which is very deep compared to usual CNNs/ResNets (considering, e.g. K=3 and D in the hundreds for images).
In particular, CNNs are typically used when there is some relevant inductive bias such as equivariance
to translations (and invariance with pooling operations) to take advantage of,
so removing this inductive bias by approximating fully-connected architectures seems a bit twisted.
The approach of reducing the function class to be approximated would seem more relevant here,
as in the cited papers Petersen & Voigtlaender (2018) and Yarotsky (2018), and perhaps the results of
the present paper can be useful in such a scenario as well.

Separately, the presentation of the paper could be significantly improved,
for instance by introducing relevant notions more clearly in the introduction and related work sections,
and by providing more insight and discussion of the obtained results in the main paper.

More specific comments:
- Section 1, p.2: define M? define D? M seems to be used for different things in different paragraphs
- Section 2: Explain what is "s" in the Barron class, or at least point to the relevant definition in the paper
- Section 3.1:
  * 'estimation error' is usually called '(expected) risk' in the statistical literature (also in the introduction). estimation error would have to do with relating R and R^hat
  * why is the estimator "regularized"?
- Definition 2: shouldn't it be D_m^(0) = D instead of 1?
- Theorem 1: What is L? Also, it would be helpful to sketch the construction in the main paper given that this is the main result.
- Section 4.2: M_1 is the Lipschitz constant of what function?
- Section 5.1: "M = 1" this is confusing, maybe use a different letter for the ridge expansion? The discussion on 'relative scale' could be made clearer.
- Section 5.2, 'if we carefully look at their proofs': more details on this should be provided.

---

> ### Author Response · Authors · 2018-11-14
> **Reply from authors (1/2)**
>
> Thank you for your detailed review. We would appreciate your insightful comments. We reply to your comments one by one.
>
> > However, the obtained CNN approximating architectures look quite unrealistic compared to most practical use-cases of CNNs,
> > since they specifically try to reproduce a fully-connected architecture, leading to residual blocks of depth ~= D/K,
> > which is very deep compared to usual CNNs/ResNets (considering, e.g. K=3 and D in the hundreds for images).
>
> It is true that the residual blocks in the original ResNet have 2 layers, while those in ours have much more layers. However, identity connections skipping many layers are not rare. For example, one of the variants of DenseNet (Huang (2017)) used to train ImageNet consists of 201 layers, and its outermost connection skips 48 layers (>20% of the whole networks).
>
> Although It might be a different discussion point,  in view of sparsity, we would argue that our NNs are more practical than ones used in previous literature (e.g., Yarotsky (2017), Schmidt-Hieber (2017), and Imaizumi & Fukumizu (2018)). They imposed somewhat artificial sparse constraints to FNNs by restricting the number of non-zero parameters. However, we need to train with L0 regularization to realize such NNs and hence actual NNs do not have such non-zero sparsity patterns. Contrary to that, our CNN is dense in general since the block-sparse FNNs from which we construct CNNs have dense blocks. We have fixed our paper to remove the sparsity constraints to made it clear that our CNN is dense in general.
>
>
> > In particular, CNNs are typically used when there is some relevant inductive bias such as equivariance
> > to translations (and invariance with pooling operations) to take advantage of,
> > so removing this inductive bias by approximating fully-connected architectures seems a bit twisted.
>
> As appeared in Zhou (2018) or Petersen & Voigtlaender (2018), it is one of the standard approaches in the function approximation theory for CNNs to approximate a target function with FNNs and to transform the FNNs into CNNs. Although this approach is somewhat indirect as you pointed out, we believe it is still useful from a viewpoint of inductive bias, too. If we can successfully reflect inductive biases as particular structures of FNNs, like block-sparseness as we did in this paper, CNNs can capture the biases via FNNs. Although this is just an idea, if the dataset has some invariance (such as translation invariance), we can expect blocks in an FNN might have some redundancy in some sense (e.g., blocks are similar to each other). Using the weight-sharing property of CNNs, we might need fewer parameters to realize a function using CNNs than using FNNs, as we pointed out in the conclusion section.
>
>
> > Separately, the presentation of the paper could be significantly improved, for instance by introducing relevant notions more clearly in the introduction and related work sections, and by providing more insight and discussion of the obtained results in the main paper.
>
> Thank you for your suggestion. We are thinking to add an extended discussion in the next revision.

---

> ### Author Response · Authors · 2018-11-14
> **Reply from authors (2/2)**
>
> Reply to specific comments:
> > Section 1, p.2: define M? define D? M seems to be used for different things in different paragraphs. The discussion on 'relative scale' could be made clearer.
>
> We added the definition of D and M to the introduction section. We used the variable M mainly for three meanings: the number of blocks of a block-sparse FNN, the number of residual blocks in a ResNet-type CNN, and the number of parameters of an NN (either FNN or CNN). As we see from Theorem 1, the CNN which we constructed an FNN with M blocks has M residual blocks (plus the 0-th block). Therefore, we used the same character M. Since an FNN with M blocks has \tilde{O}(M) parameters in common settings, we used M to indicate the number of parameters in the introduction. If it is confusing, we are thinking to use different characters for parameter counts and block counts.
>
>
> > Section 5.1: "M = 1" this is confusing, maybe use a different letter for the ridge expansion?
>
> “M=1” should have been D_1 = \cdots = D_M = 1 and L_1^{(1)} = \cdots = L_M^{(1)} = 1. We have fixed the description of Section 5.1 and Section E.
>
> > Section 2: Explain what is "s" in the Barron class, or at least point to the relevant definition in the paper
>
> “s” is a parameter in the definition of the Barron class that indicates the decay speed of signals in Fourier domain. We have added the reference to Definition 3.
>
> > Section 3.1:
> >   * 'estimation error' is usually called '(expected) risk' in the statistical literature (also in the introduction). estimation error would have to do with relating R and R^hat
>
> Indeed, in the statistics literature, the estimation error is frequently used for the (finite-dimensional) parameters. On the other hand, in nonparametric statistics, it is also common to use the terminology "estimation error" to indicate the expected risk, because parameters themselves are functions in the L2-space (while the estimation error is also sometimes referred as the variance term inside a model). Therefore, we used the terminology.
>
> >   * why is the estimator "regularized"?
>
> We called this estimator "regularized" because we impose sparse constraints on the set of CNNs from which we pick the ERM estimator by restricting the maximum number of non-zero parameters. Now we do not impose such constraints, we have replaced it with the clipped ERM estimator.
>
> > Definition 2: shouldn't it be D_m^(0) = D instead of 1?
>
> Yes. Thank you for pointing it out. We have fixed it.
>
> > Theorem 1: What is L? Also, it would be helpful to sketch the construction in the main paper given that this is the main result.
>
> We intended that L is the total depth of the ResNet-type CNNs. We have changed the statement to specify the ResNet-type CNNs by the number of residual blocks and the depth of each block as we did in the definition of \mathcal{F}^{\mathrm{(CNN)}}.
>
> > Section 4.2: M_1 is the Lipschitz constant of what function?
>
> The Lipschitz constant of a function realized by a CNN in \mathcal{F}^{\mathrm{(CNN)}}.
>
> > Section 5.2, 'if we carefully look at their proofs': more details on this should be provided.
>
> We have added the detail of the proof as Lemma 7.

---

### Official Review · AnonReviewer2 · 2018-11-03
**Approximate block sparse fully connected neural networks, the Barron class and the Holder class using the Residual CNNs**

**Rating:** 6
**Confidence:** 3

**Review:**

The authors demonstrate the function expression properties for the Residual type convolutional neural networks to approximate the block sparse fully connected neural networks. Then it is shown that such Res-CNNs can approximate any function as long as it can be expressed by the block-sparse FNNs, including the Barron class and Holder class functions. The price to pay is that the number of parameters is larger than that of the FNNs by a constant factor.

The idea for connecting the expressive ability of CNNs with FNNs is interesting, which can fully take advantage of the power of FNNs to understand CNNs. However, it is not very clear how the convolutional structure of CNNs help in the analysis of approximating FNNs. For example, in the analysis of C.1 and C.2, it will help better understand why CNNs may work from a high-level intuition when the authors construct the filters.

Moreover, it will also help better understand the expressive power of CNNs if the authors can provide some extended discussion on why approximating the block-sparse FNNs rather than arbitrary feed-forward networks. Is there any fundamental reason (or a counterexample) this cannot be realized, or is there to some extent a technical barrier in the analysis?

Minor issue

On page 20, “Bounds residual blocks” -> “Bounds for residual blocks”

---

> ### Author Response · Authors · 2018-11-14
> **Reply from authors**
>
> We appreciate your detailed and insightful comments. We reply to your comments one by one.
>
> > However, it is not very clear how the convolutional structure of CNNs help in the analysis of approximating FNNs. For example, in the analysis of C.1 and C.2, it will help better understand why CNNs may work from a high-level intuition when the authors construct the filters.
>
> Convolution with a size-1 filter, inspired by a 1x1 convolution used in image recognition models such as Inception (Szegedy et al. (2014)), is equivalent to dimension-wise affine transformation. Intuitively, it implies CNNs have as powerful learning ability as FNNs. There is room for discussion if our proofs can effectively utilize the convolutional structure of CNNs. However, we have shown that approximation and estimation error rates are no worse than that of FNNs. In particular, CNNs can already achieve the minimax optimal rate for the H\”older class. That means even if we make full use of convolutional structure, we have no hope to improve the rate. Considering that the learning ability of CNNs had not been investigated deeply in the literature, we believe our analysis is a critical first step toward unveiling the learning ability of CNNs.
>
> With that being said, we also want to leverage the inductive bias of data to yield advantageous learning ability of CNNs over FNNs. We believe the analysis of CNNs employing FNNs could be a promising strategy. If we can successfully reflect inductive biases as particular structures of FNNs, like block-sparseness as we did in this paper, CNNs can capture the biases via FNNs. Although this is just an idea, if the dataset has some invariance (such as translation invariance), we could expect blocks in an FNN has redundancy in some sense (e.g., blocks are similar to each other). Using the weight-sharing property of CNNs, we might need fewer parameters to realize a function using CNNs than using FNNs, as we pointed out in the conclusion section.
>
>
> > Moreover, it will also help better understand the expressive power of CNNs if the authors can provide some extended discussion on why approximating the block-sparse FNNs rather than arbitrary feed-forward networks. Is there any fundamental reason (or a counterexample) this cannot be realized, or is there to some extent a technical barrier in the analysis?
>
> Almost all FNNs used in the previous studies to approximate some specific function classes have block-sparse structures. For example, Yarotsky (2017), Yarotsky (2018), and Zhou (2018). Same is true of the case of the expansion of a Besov function by wavelet bases (Bölcskei et al. (2017). From the viewpoint of functional analysis, block-sparse structure naturally corresponds to expansion of the target function with a set of basis functions approximated by dense FNNs.
> It is not trivial how to (approximately) transform general FNNs without block-sparse structures into ResNet-type CNNs, because there is no principled way to decompose an FNN into residual blocks. Although block-sparse FNNs are somewhat theoretical tools, we can realize optimal dense ResNet CNNs, by bypassing them.

---

> > ### Comment · AnonReviewer2 · 2018-11-22
> > **Thanks for the update**
> >
> > I do not have further questions.

---

### Official Review · AnonReviewer3 · 2018-11-04
**The block sparse structure seems unnecessary given the results in [Schmidt-Hieber 2017].**

**Rating:** 4
**Confidence:** 4

**Review:**

This manuscript shows the statistical error of the ERM for nonparametric regression using the family of a Resnet-type of CNNs. Specifically, two results are showed. First, the authors show that any block-sparse fully connected neural network can be embedded in CNNs. Second, they show the covering number of the family of CNNs. Combining with the existing results of the approximation error of neural nets (Klusowski&Barron 2016, Yarotsky 2017, Schmidt-Hieber 2017), they show the L2 statistical risk.

Detailed comments:

1. The intuition of using block-sparse FNN seems unclear. It seems that when $M=1$, it reduces to the sparse NN considered in [Schmidt-Hieber 2017]. In the proof of Corollary 5, the authors directly use the error of approximating Holder smooth function by sparse FNN and show that the construction in [Schmidt-Hieber 2017] is actually block-sparse. Thus, it seems unclear why we should consider such block-sparse family. Can any sparse NN be embedded in the family of CNNs?

2. In the Related Work, the authors only compare with 2 previous work on the approximation error of CNN. Actually, this work is more related to [Schmidt-Hieber 2017] due to borrowing the results. It would be better to see what the novelties are compared with that work, especially in terms of the proof techniques.

3. The authors claim that the construction of approximator for Holder functions in [Schmidt-Hieber 2017] is block sparse. It would be nice to give more details of the construction since this is not claimed in [Schmidt-Hieber 2017].

---

> ### Author Response · Authors · 2018-11-14
> **Reply from authors**
>
> Thank you for your review. We appreciate your detailed feedback. We reply to your comments one by one.
>
> 1.
> > It seems that when $M=1$, it reduces to the sparse NN considered in [Schmidt-Hieber 2017].
> Blocks in a block-sparse FNN is dense in general, as opposed to the sparse NN. Therefore, a block-sparse FNN with M=1 block is different from NNs used in [Schmidt-Hieber 2017]. Of course, we can apply our theorem with M=1 to derive the estimation error. The resulting CNN would be a ResNet with single residual block (since the number of blocks in FNN equals to the number of residual blocks in transformed ResNet-type CNN). However, the CNN has as many as O(M) channels. Since optimal FNN has M=O(N^\alpha) (\alpha > 0) blocks, we cannot keep the number of units per layer of the optimal CNN constant.
>
> > Thus, it seems unclear why we should consider such block-sparse family. Can any sparse NN be embedded in the family of CNNs?
> Almost all FNNs used in the previous studies to approximate some specific function classes have block-sparse structures. For example, [Yarotsky 2017], [Yarotsky 2018], and [Zhou 2018]. Same is true of the case of the expansion of a Besov function by wavelet bases [Bölcskei et al. 2017]. From the viewpoint of functional analysis, block-sparse structure naturally corresponds to expansion of the target function with a set of basis functions approximated by dense FNNs.
> in view of sparsity, we would argue that our NNs are more practical than ones used in previous literature (e.g., [Yarotsky 2017], [Schmidt-Hieber 2017], and [Imaizumi & Fukumizu 2018]). They imposed somewhat artificial sparse constraints to FNNs by restricting the number of non-zero parameters. However, we need to train with L0 regularization to realize such NNs and hence actual NNs do not have such non-zero sparsity patterns. Contrary to that, our CNN is dense in general since the block-sparse FNNs from which we construct CNNs have dense blocks. We have fixed our paper to remove the sparsity constraints to made it clear that our CNN is dense in general.
>
> 2.
> > In the Related Work, the authors only compare with 2 previous work on the approximation error of CNN. Actually, this work is more related to [Schmidt-Hieber 2017] due to borrowing the results.
> We compared our work with Zhou (2018) and Petersen & Voigtlaender (2018) since their works are close to ours in analyzing approximation ability (and hence estimation ability) of CNNs.
>
> > It would be better to see what the novelties are compared with that work, especially in terms of the proof techniques.
> We think the evaluation of the covering number of the set of CNNs is novel. It corresponds to the evaluation of M_1 in Theorem 2. If we naively trace the proof of [Schmidt-Hieber 2017] Lemma 12, the logarithm of covering number is prohibitively large to derive the desired estimation error.
> We mainly used two techniques to deal with the problem: the architecture-aware evaluation of sup norms and the rescaling of parameters. First, we explicitly used the ResNet-type architecture and derived a tighter bound of the sup norms of the function realized by a CNN (Proposition 11 and Lemma 3). Secondly, if we apply our result on estimation error bounds to concrete classes using the approximation results of [Klusowski 2018] (for Barron class) and [Schimidt-Hieber 2017] (for H\”older class), the assumption of Corollary 1 about M_1 is not satisfied because the Lipschitz constant of the CNN is too large. We devised the parameter rescaling technique to reduce the Lipschitz constant to meet the assumption. We discussed the problem in Section 5.1 (Barron class), Section 5.2 (H\”older class), and Lemma 6.
>
> 3. We have added the detail proof as Lemma 7.

---

### Author Response · Authors · 2018-11-14
**Revised version uploaded**

We have uploaded the revised version of our paper. The main differences from the previous one are as follow:

- We removed the sparsity constraints (specified S by the previous version) from $\mathcal{F}^{\mathrm{(CNN)}}$ in order to emphasize that the CNNs we consider is dense in general. Accordingly, the statements of Theorem 2 and Corollary 1 (and Lemma 4) are changed so that they do not use S.
- We added the lemma (Lemma 7) on how to approximate the \beta-H\”older function using block-sparse FNNs by modifying the proof of Schmidt-Hieber (2017).
- Fixed typos and grammatical errors and changed several variables for readability.

Thank you for your interest.

---

### Meta-Review · Area_Chair1 · 2018-12-17
**Interesting transformation of block-sparse fully connected net to ResNet Convolutional blocks, yet the ResNet architecture seems unrealistic and indirect.**

**Confidence:** 4
**Recommendation:** Reject

**Metareview:**

The paper presents an interesting treatment of transforming a block-sparse fully connected neural networks to a ResNet-type Convolutional Network. Equipped with recent development on approximations of function classes (Barron, Holder) via block-sparse fully connected networks in the optimal rates, this enables us to show the equivalent power of ResNet Convolutional Nets.

The major weakness in this treatment lies in that the ResNet architecture for realizing the block-sparse fully connected nets is unrealistic. It originates from the recent developments in approximation theory that transforming a fully connected net into a convolutional net via Toeplitz matrix (operator) factorizations. However the convolutional nets or ResNets obtained in this way is different to what have been used successfully in applications. Some special properties associated with convolutions, e.g. translation invariance and local deformation stability, are not natural in original fully connected nets and might be indirect after such a treatment.

The presentation of the paper is better polished further. Based on ratings of reviewers, the current version of the paper is on borderline lean reject.